# Bioinspired adaptable multiplanar mechano-vibrotactile haptic system

Sara-Adela Abad [1,2] ✉, Nicolas Herzig [3], Duncan Raitt [1], Martin Koltzenburg [4] & Helge Wurdemann [1]

Several gaps persist in haptic device development due to the multifaceted nature of the sense of touch. Existing gaps include challenges enhancing touch feedback fidelity, providing diverse haptic sensations, and ensuring wearability for delivering tactile stimuli to the fingertips. Here, we introduce the Bioinspired Adaptable Multiplanar Haptic system, offering mechanotactile/steady and vibrotactile pulse stimuli with adjustable intensity (up to 298.1 mN) and frequencies (up to 130 Hz). This system can deliver simultaneous stimuli across multiple fingertip areas. The paper includes a full characterisation of our system. As the device can play an important role in further understanding human touch, we performed human stimuli sensitivity and differentiation experiments to evaluate the capability of delivering mechano-vibrotactile, variable intensity, simultaneous, multiplanar and operator agnostic stimuli. Our system promises to accelerate the development of touch perception devices, providing painless, operator-independent data crucial for researching and diagnosing touch-related disorders.

The sense of touch is key to perceiving and interacting with the environment. This sense is used to differentiate characteristics of objects (e.g., textures, rigidity, plasticity, and weight) and clues (e.g., vibrations for determining heartbeat or localising arteries). In areas like medicine, remote control systems, entertainment, and training, the use of touch feedback technologies is increasing. It helps users feel like they are interacting with physical objects, even if they are far away or only exist in a virtual world. This could be anything from animals or even a cancerous organ. The goal is to help people experience rare or inaccessible situations (e.g., interacting with wild or underwater animals or being trained to diagnose rare conditions), widen their knowledge and work more accurately and safely, especially when performing demanding tasks (e.g., palpation). Furthermore, touch is also studied for its important role in social relationships[1], in strengthening bonds between people[2].

According to the work by Jones[3], the human skin contains myelinated $A\beta$ fibres that respond to mechanical stimuli with the intensity of the stimuli being correlated to their discharge frequency[3] (the frequencies are summarized in Fig. 1a). These fibres end in Merkel, Meissner, Pacinian, or Ruffini corpuscles. The Merkel-SA1 (Slow Adapting type I) corpuscles in the finger are sensitive to steady force, low frequency ($f < 5$ Hz), dynamic skin deformation[4], and local spatial discontinuities. They have a higher sensitivity to surface features and curvatures. The Meissner-FAI (Fast Adapting type I) corpuscles are four times more sensitive to dynamic skin deformation/motion than the SA1 corpuscles. They can detect sudden forces associated with hand-held objects and are responsive to pressures and vibrations between 5 to 50 Hz[4]. The Ruffini-SAII (Slow Adapting type II) corpuscles provide information about the direction of motion or force, particularly when the motion involves skin stretching. They are sensitive to skin stretch and steady forces. The Pacinian-FAII (Fast adapting type II) corpuscles are made to capture large low-frequency stresses and strains encountered in daily manual activities. They have a very low spatial resolution, and respond to distant stimuli[4]. FAII are sensitive to micrometric deformations and vibrotactile stimuli in the range of 40 Hz to > 400 Hz[4-6]. Moreover, as stimuli intensity increases, there is

[1]Department of Mechanical Engineering, University College London, London, UK. [2]Faculty of Agriculture and Renewable Natural Resources, Universidad Nacional de Loja, Loja, Ecuador. [3]School of Engineering and Informatics, University of Sussex, Brighton, UK. [4]Queen Square Institute of Neurology, University College London, London, UK. ✉e-mail: s.abad-guaman@ucl.ac.uk

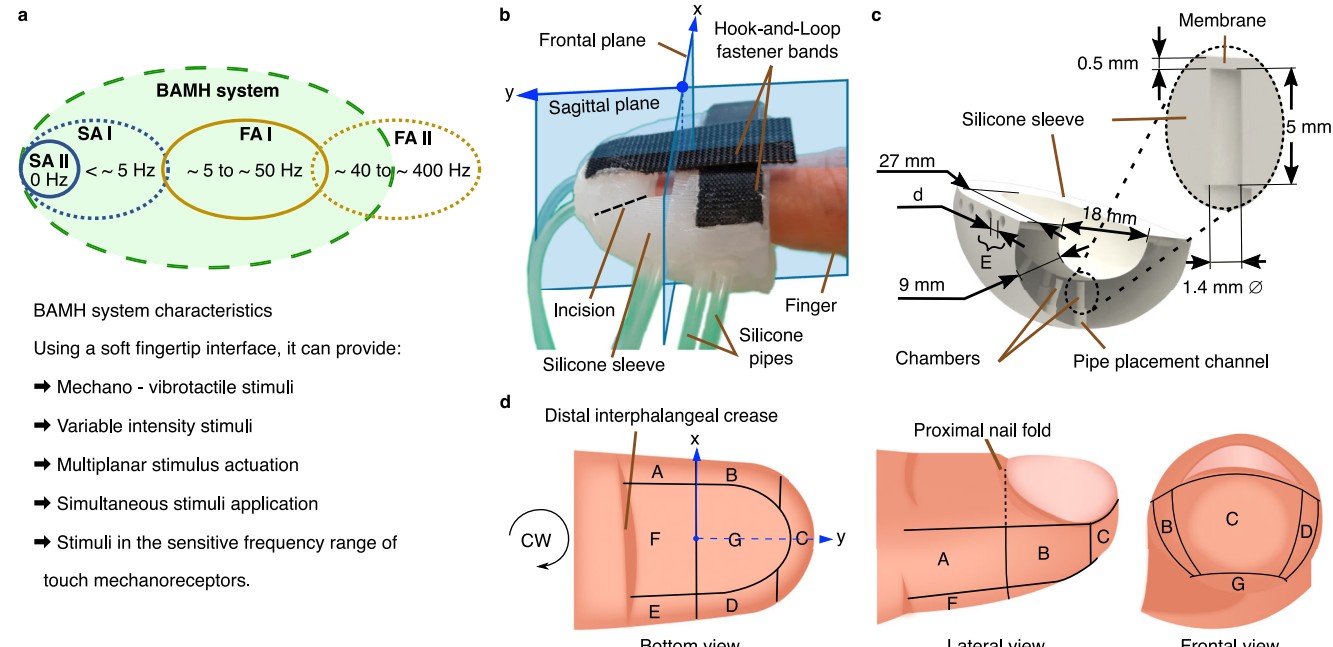

**Fig. 1 | Haptic fingertip interface. a** Characteristics of the stimuli provided by our Bioinspired Adaptable Multiplanar mechano-vibrotactile Haptic (BAMH) system through the fingertip interface, where SA and FA refer to the slow and fast adapting corpuscles, respectively. **b** The main components of the soft compliant fingertip interface are the silicone sleeve, the pneumatic pipes for increasing the internal pressure of the chambers, and the Hook-and-Loop fastener bands for tightening the interface to the finger. **c** Fingertip interface dimensions and internal structure, where E is a lateral area exhibiting the two chambers per area and the stimuli distance *d* is the edge-to-edge distance between the area's chambers. **d** The seven areas are defined using clockwise (CW) direction and the distal phalanx natural landmarks, e.g., the distal interphalangeal crease and the proximal nail fold.

a corresponding increase in the FAII peak response at lower frequencies. To clarify, when the stimulus induces a peak-to-peak skin displacement of ≥6 $\mu$m at a frequency of ≤128 Hz, it triggers a higher FAII peak response compared to frequencies exceeding ≤128 Hz[5].

Therefore, when fingers interact with the environment (e.g., during grabbing, pinching, or palpation), humans use different areas/planes of the fingers' distal phalanx. This may be explained by the different receptive fields and density distribution of the four mechanoreceptors across the finger. So, to stimulate the mechanoreceptors responsible for touch sensing in human skin, haptic interfaces should adapt to the curvature of the fingers and be capable of providing simultaneous multiplanar stimuli that change in intensity and frequency. The latter should be from 0 Hz to higher than 50 Hz (see Fig. 1a).

Visual feedback is a low-cost solution that displays the contact information graphically. For example, tactile cues are overlaid on the camera image of a laparoscope[7]. Understanding this information partially overlaid over real images requires training so users can quickly connect what they see with the related touch feedback information.

Kinesthetic feedback integrates instrument palpation methods using direct force feedback. It can require the implementation of bilateral control schemes into the existing control architecture by adding sensing devices on the worker side and actuators on the manager side[8]. Some of the challenges include system volume[9,10] and the trade-off between force feedback transparency and system stability (position/force feedback control)[11]. To improve manufacturability and provide more comprehensive haptic cues in a relatively small volume, origami-based robots such as Foldaway[12] and FingerPrint[13] have been proposed. However, further development is needed to provide multiplanar and simultaneous feedback.

Haptic tactile feedback provides cutaneous stimuli by using different technologies and techniques. Wearable tactile actuators/hand exoskeletons comprise technologies such as rigid tactile pin displays[14–16], soft vibrotactiles[17], and focused ultrasound[18]. The techniques explored to achieve tactile sensation include pneumatic air pressure applied to the skin of the fingertips[19], inflatable tactile cells integrated and validated into a surgical robot[20], and a combination of tactile pins and kinesthetic feedback for palpation simulators[14]. However, a number of gaps persist in the area of haptic device development due to the multifaceted nature of the sense of touch, influenced by various factors such as social context, temperature, shear force exertion, and characteristics of tactile stimuli[6,21–23]. Existing gaps include[24,25] challenges enhancing levels of realism and fidelity in haptic devices, integrating multimodal sensations effectively, encompassing mechanotactile/steady and vibrotactile feedback, or temperature variations, and miniaturising both the overall interface and its actuators to minimise user distraction and optimise focus on the tactile experience[24,25]. Finding solutions to these challenges is essential for advancing haptic technologies and unlocking their full potential across various applications.

Here, we show that our work contributes to enhancing the fidelity of touch feedback, providing a frequency range of haptic mechanotactile feedback sensation, and allowing wearability. In particular, the innovation of our Bioinspired Adaptable Multiplanar Haptic (BAMH) system lies in a device that:

1. is able to provide both mechanotactile/steady as well as vibrotactile pulse stimuli with variable intensity over a wide range of frequencies. Hence, the device can stimulate mechanoreceptors including the SAII (sensitive to steady stimulus), the SAI (sensitive to a vibrotactile stimulus with a frequency lower than 5 Hz), the FAI (sensitive to vibrotactile stimulus with a frequency between 5 Hz and 50 Hz), and the FAII (sensitive to vibrotactile stimulus with a frequency between 40 Hz and 400 Hz).

2. is capable of provide simultaneous stimuli on several planar areas of the entire fingertip surface, i.e., the frontal, lateral, and bottom areas of the finger.

The stimulus' intensity range and vibrotactile pulse frequency range are evaluated through the characterisation of the BAMH system. In addition, we performed human stimuli sensitivity and differentiation experiments to evaluate the capability of the BAMH system to deliver mechano-vibrotactile, variable intensity, simultaneous, multiplanar, and operator agnostic stimuli, as we believe that our system can play an important role in further understanding human touch.

## Results and Discussion

### BAMH system

The Bioinspired Adaptable Multiplanar mechano-vibrotactile Haptic (BAMH) system is a pneumatically actuated, soft-material robotic interface. The combination of a soft material, silicone-based structure with air actuation offers a number of benefits:

- For fluidic actuation, an extrinsic method, any pressure regulators and supply of pressurized fluid is separated from the haptic feedback device itself[13], hence, allowing wearability of the haptic device[24].
- In particular, pneumatic actuation offers a lighter alternative to hydraulic actuation, thereby enhancing the wearable nature of the system[24].
- Soft material, pneumatically actuated robots are inherently regarded as safe[26], thus enabling the delivery of painless mechanotactile stimuli to the skin through the inflation of a soft membrane with low pressure (≤150 kPa).

- Implementation of a soft-materials approach facilitates redesign of moulds to manufacture personalized devices[27] and, hence, enhances adaptation to the contours of the fingertip or other body parts[28]. On the other hand, soft material properties, such as hardness, allow further modification of the characteristics of the provided haptic stimuli depending on the applications[29].

The BAMH system comprises the electro-pneumatic control subsystem and the soft fingertip interface.

The control subsystem, illustrated in Fig. 2, comprises electric and pneumatic components. This subsystem design allows (i) providing steady/mechanotactile stimulus and vibrotactile pulse stimulus, (ii) generating vibrotactile pulse stimulus with a maximum built frequency of 280 Hz (defined by the fast switching valves[30]) to stimulate the four main mechanoreceptors (slow and fast adapting type I and II) related to human touch perception, (iii) changing the stimuli intensity by providing a pressure output from 0 kPa to 300 kPa, (iv) stimulating simultaneously all the seven areas of the finger, and (v) setting a different stimuli characteristics for each chamber (see Methods section - Haptic feedback system components description and characterisation).

The soft fingertip interface comprises a silicone sleeve, internal chambers, and Hook-and-Loop fastener bands that allow the haptic interface to be tightened to the finger (see Fig. 1b, c). The fingertip contains fourteen internal chambers distributed in pairs across seven

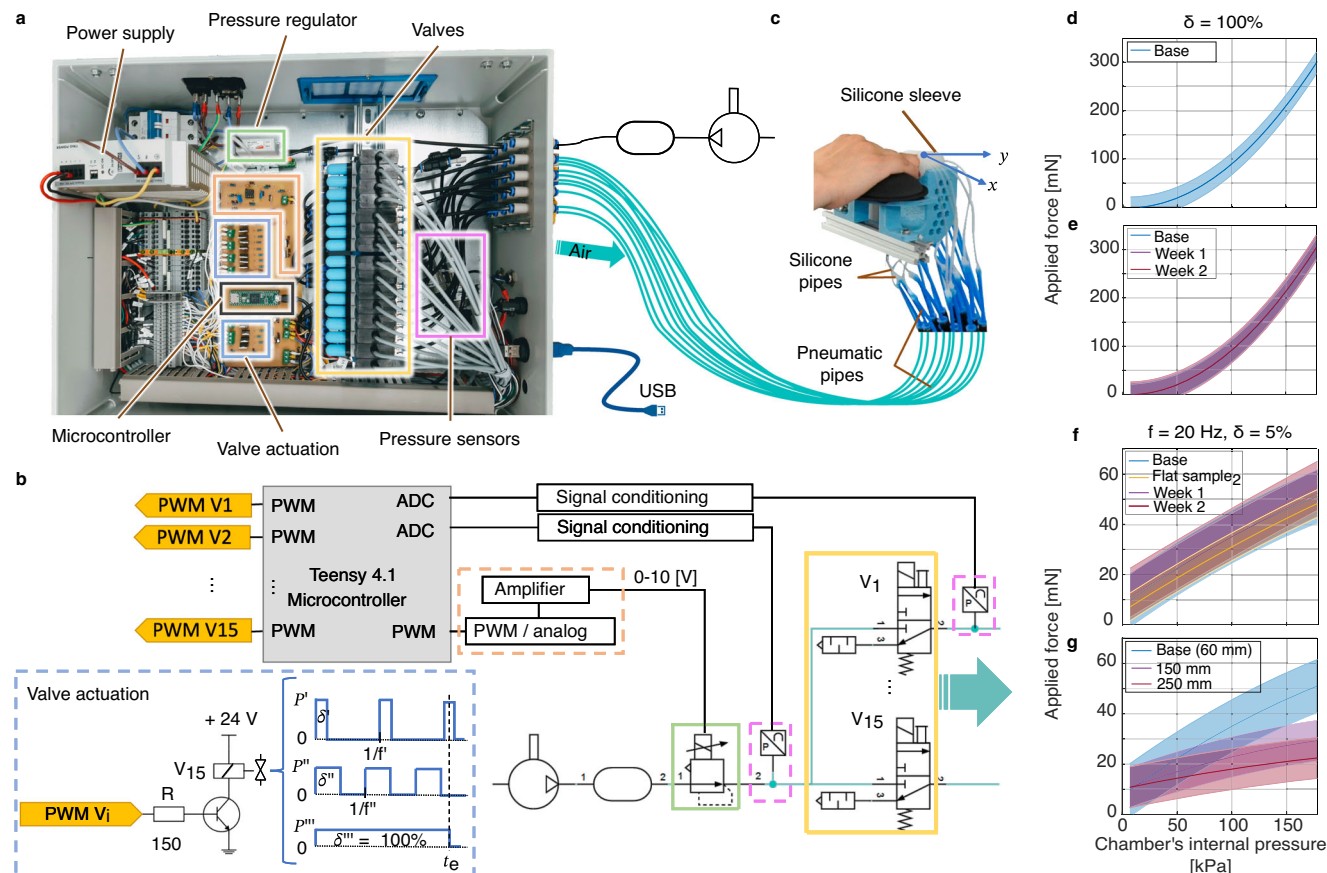

**Fig. 2 | Bioinspired Adaptable Multiplanar mechano-vibrotactileHaptic (BAMH) system.** It comprises the control subsystem and the fingertip interface. **a** The control subsystem and **b** its electro-pneumatic diagram. The pressure regulator (green rectangles) defines the stimuli intensity. The pressure sensors (magenta rectangles) monitor the air supply and chamber's pressures. The signal conditioning circuits (orange rectangles) scale the signals received and sent by the microcontroller. A valve actuation circuit(blue rectangles) generates actuation signals with different duty cycle δ, intensities, and frequencies *f*, illustrating three representative signals. Each valve is connected to a chamber of **c** the fingertip interface. The chamber's internal pressure-force relationship **d** for steady stimulus and **e** its performance over time; and for vibrotactile pulse (*f* = 20 Hz and δ = 5%) stimulus **f** over time and two flat samples and **g** for different pipe lengths. Base refers to flat sample 1, day 1, and 60 mm pipe length.

**Table 1 | BAMH's system results compared to literature's haptic feedback systems that provide tactile feedback**

| Haptic systems | Haptic interface material | Mechano-Vibrotactile stimulus | Freq. Range Hz | Max. Force mN | Areas | | | Simultaneous stimuli across areas |
|---|---|---|---|---|---|---|---|---|
| | | | | | Lat | Bot | Frnt | |
| Our BAMH system | Dragon Skin™ 20 | M, V | 0–130 | 298.1 | Y | Y | Y | Y |
| Pin display[16] | Rigid | M** | – | 400 | Y | Y | N | Y |
| SPA-skin[17] | Sylgard 184 | M*, V | 0–100 | 1000 | N | Y | N | N |
| SPA-skin[23] | Dragon Skin™ 30 | M*, V | 0–120 | 300 | N | Y | N | N |
| Fuppeteer[49] | Rigid | M | – | 2100 | Y | Y | Y | N |
| FingerPrint[13] | Flexible 80A | M*, V | 1–64 | 7000 | N | Y | N | N |
| HAXEL[50] | Various flexible | M, V | ~ 10–320 | 300 | N | Y | N | N |

M and V denotes mechano and vibro tactile stimulus, respectively.

* Denotes that it is our interpretation based on the information provided in the corresponding paper.

** In this work, only steady/mechano tactile feedback is used. Nevertheless, the system has an update time of 50 Hz, so it may be capable of delivering mechanovibrotactile stimuli with a frequency lower than 50 Hz.

areas defined using natural landmarks, such as the proximal nail fold and the distal interphalangeal crease (see Fig. 1d). Each chamber is airtight and connected to the output of fast-switching valves. This connection allows actuation of the 0.5 mm chamber membrane exhibited in Fig. 1c. The fingertip interface is built using a moulding approach. For the application of measuring human stimuli sensitivity and differentiation capabilities, Dragon Skin™ 20 Smooth On was used as material (see Supplementary Fig. 1 and Methods). Using the same manufacturing process, four different fingertip interfaces were built to evaluate human differentiation. They differ in the internal edge-to-edge chamber distance, $d$, for each chamber pair. $d$ was kept constant across areas but changed across fingertip interfaces. $d$ for the four fingertip interfaces are 2 mm, 3 mm, 4 mm, and 5 mm. Due to the softness of the bioinspired fingertip interface and the location of the chambers across the seven areas of the finger, this interface can adapt to the curvature of human fingers, and the system can stimulate the finger on several planes.

The selection of Dragon Skin™ 20 Smooth On for the fingertip interface was driven by the material properties' impact on tactile stimuli and the need to accommodate human sensitivity thresholds. Its softness enhances adaptation to finger curvature, while its lower internal pressure requirement compared to other materials facilitates stimulation. A 0.5 mm membrane thickness was chosen by the empirical experience of our manufacturing methods. It is worth noting that a thicker membrane would require higher internal pressure to inflate and stimulate the skin. Based on the results we obtained through manual stimuli sensitivity and two-point differentiation experiments, our haptic feedback devices required a chamber diameter of less than 2 mm. Hence, a diameter of 1.4 mm was feasible considering similar haptic interfaces[20,31].

## BAMH system characterisation

The haptic system's stimuli characterisation defines the relationship between the chamber's internal pressure and the force it applies. We used flat samples, whose fabrication follows a similar procedure employed for fabricating the fingertip interface.

For steady/mechanotactile stimulus characterisation, when loading (incrementing the chamber's internal pressure), see Fig. 2d, the minimum average force applied is 1.72 mN. The maximum average force applied is 298.1 mN, with a hysteresis of 17.21 mN, and an average resolution of 1.57 mN. This training force data, collected on day 1, were fitted to a second-degree polynomial. The difference between the data calculated using the polynomial and the acquired testing data (root mean square deviation-RMSD) was 11.07 mN. When evaluating the performance of the system over time (see Fig. 2e), the difference between the base (flat sample 1, day 1, second-degree polynomial data) and its week 2 acquired testing data was 11.58 mN.

However, across two different flat samples, flat sample 2 - day 1 test data comparison to base data revealed a difference of 21.74 mN.

For vibrotactile pulse stimulus characterisation, Fig. 2f and Supplementary Table 1 show that when the week 2 test data was plotted against the corresponding base data, the difference (RMSD) was 6.23 mN. Furthermore, the flat sample 2 test data compared to base data produces an RMSD of 6.46 mN. The applied force for the same internal chamber pressure is also lower than that corresponding to a steady stimulus ($\delta = 100\%$). As illustrated in Fig. 2g, for a 60 mm pipe length, the force range is up to 50.65 mN, RMSD is 5.64 mN, and hysteresis is 4.74 mN. The results also highlight that increasing the pipe length from 60 mm to 250 mm decreased the force range from 50.65 mN to 22.29 mN. Analysis showed a significant correlation of $\delta$ and pipe length on max frequency (multiple regression ANOVA, $F(2, 57) = 142.20$, $p < 0.001$, $R^2 = 0.83$, Adjusted $R^2 = 0.83$). Modulated tests showed little change in max frequency with pipe length (multiple regression, $\beta = -0.04$, $T(60) = -1.40$, $p = 0.17$, see Supplementary Table 2). However, $\delta$ is significantly correlated with max frequency (multiple regression, $\beta = 1.35$, $T(60) = 16.8$, $p < 0.001$). Duty cycles, $\delta$, of 5%, 10%, 25%, 50%, and 75% corresponded to max usable frequencies of 25 Hz, 55 Hz, 90 Hz, 95 Hz and 130 Hz, respectively. Additionally, a variation in $\delta$ reflected a variation in the intensity of the stimulus. This suggests that higher duty cycles may be needed for higher actuation frequencies. Supplementary Figs. 2–5 illustrate the raw force data of mechanotactile and vibrotactile pulse stimulus delivered by the BAMH system. The spectral coefficients at the frequencies that are a multiple of the pulse stimulus are evidence of the BAMH's system capability to deliver pulse stimulus. Due to inherent limitations in the frequency analysis of real noisy data (e.g., the evaluated time series not containing complete periods of the pulse, see Methods), the magnitude of the spectral components is not considered in the analysis.

The characterisation of the haptic system demonstrates that the system can provide variable intensity as well as multiplanar and simultaneous mechano-vibrotactile stimuli to the distal phalanx of the fingers. Furthermore, the stimuli intensity can be changed through the pressure regulator or the switching frequency and duty cycle of the actuation valve. The difference between the data calculated using the polynomial and the acquired data across flat samples suggests that an initial calibration is needed for each fingertip interface. This finding can be explained by the use of moulding for the fingertip interface manufacture. The manual steps in manufacturing can be the source of behaviour variation across fingertip interfaces.

When comparing the results of our BAMH system with existing haptic feedback systems providing tactile feedback, as outlined in Table 1, it can be observed that our system, by using Dragon Skin™ 20 as material for the soft fingertip interface can deliver mechano-

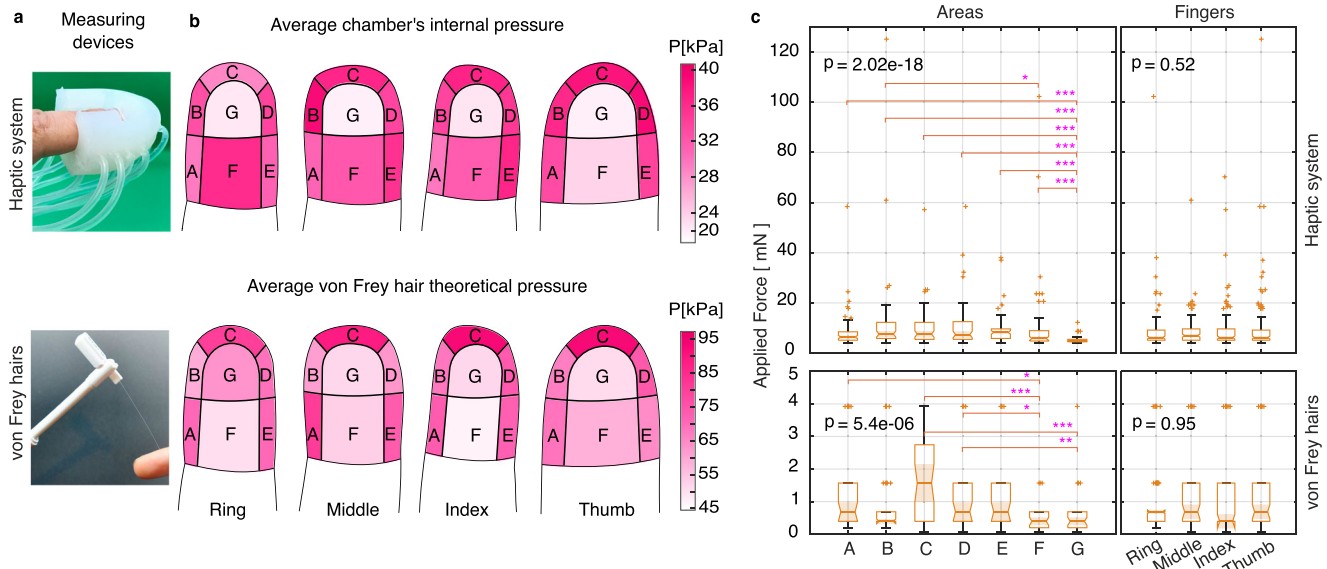

**Fig. 3 | Distal phalanx sensitivity. a** Devices used. Sensitivity across fingers and areas is illustrated as **b** the average internal pressure of the chambers (haptic system) and the average theoretical applied pressure (handheld device: von Frey hairs), where the magnitude of the pressure is linked to the color intensity of the areas. **c** Sensitivity across areas and fingers is also presented as the applied force in the finger, where the median is highlighted with a notch, + are outliers, and each box's limits are the second and third quartiles. *, **, and *** represent statistical significance with $p \leq 0.05$, $p \leq 0.01$, and $p \leq 0.001$, respectively.

vibrotactile, with a maximum pulse frequency of 130 Hz. Our system is the only one capable of stimulating both Slow Adapting mechanoreceptors (sensitive frequency range between 0 Hz and < ~ 5 Hz) and Fast Adapting mechanoreceptors (sensitive frequency range between ~5 Hz to ≥ 400 Hz) with a frequency exceeding 120 Hz. With this capability, our system may stimulate FAII mechanoreceptors at 128 Hz, wherein FAII exhibits the highest peak response when the stimulus induces ≥ 6 $\mu$m skin deformation[5]. However, the maximum force output of our system was recorded at 298.1 mN because we were focused on the application of detecting stimuli sensitivity and differentiation thresholds. So, we have capped the maximum internal pressure of our system at 178.2 kPa instead of the 300 kPa achievable by the pressure regulators. Moreover, our bioinspired fingertip interface facilitates the delivery of stimuli across the lateral, bottom, and frontal areas of the finger with simultaneous actuation across and between these areas.

We believe that our BAMH system can contribute to further understanding of the human touch. So, we investigated human fingers stimuli sensitivity and differentiation using our system and handheld devices. Consequently, von Frey hairs, which are graded calibrated filaments that exert a constant mechanical force when bent[32] and a two-point discriminator, illustrated in Fig. 3a) were used to acquire stimuli sensitivity and differentiation data, respectively.

## Distal phalanx sensitivity
Sensitivity refers to the minimal force/pressure required for the participant to feel a stimulus. This implies the higher the minimal force, the lower the sensitivity, and vice versa. We took sensitivity into account to demonstrate the haptic system's capability to provide variable-intensity mechanotactile stimulus below and above the human sensing range. Additionally, the BAMH system sensitivity results are compared to those from von Frey hairs (handheld device), which are illustrated in Fig. 3a. Figure 3b varying intensity color illustrates the sensitivity variation across fingers and areas. This figure presents sensitivity regarding the chamber's average internal pressure for the haptic interface and the average theoretical pressure provided by the manufacturer of the von Frey hairs. The average pressure was chosen because our haptic system changes the internal pressure of the

chamber to vary the stimulus intensity. However, in the remainder of the paper, sensitivity is analyzed regarding the applied force, which is the criterion commonly employed when using handheld devices.

The varying sensitivity, median force, and results acquired with the haptic system (see Fig. 3c) demonstrate that sensitivity changes across areas (Kruskal Wallis, $n = 72$, $p = 2.02 \times 10^{-18}$), but there is not enough evidence to state that sensitivity changes across fingers (Kruskal Wallis, $n = 126$, $p = 0.52$). This figure also corroborates that the most sensitive area is the pulp of the finger (area G), with a median force of 4.8 mN, and the front of the finger (area C) demonstrates low sensitivity.

The handheld device results (see Fig. 3c) corroborate the haptic system findings. Sensitivity changes across areas (Kruskal Wallis, $n = 40$, $p = 5.37 \times 10^{-6}$), but it may not change across fingers (Kruskal Wallis, $n = 70$, $p = 0.95$). The pulp of the finger (area G) is among the areas with high sensitivity, while the front of the finger (area C) presents low sensitivity.

The index finger sensitivity analysis results across frequencies and areas (see Fig. 4a) demonstrate that the sensitivity threshold varies across regions (Kruskal Wallis, $n = 84$, $p = 2.55 \times 10^{-15}$) and frequencies (Kruskal Wallis, $n = 98$, $p = 5.03 \times 10^{-46}$). The results across frequencies are skewed to the right, indicating that the highest stimulus intensity threshold occurs at 2 Hz (two-sided Wilcoxon rank sum test, 1% significance level, $p \leq 1.58 \times 10^{-11}$), with the threshold decreasing as the frequency increases. Also, Fig. 4b confirms that the sensitivity data from von Frey hairs correlates with the BAMH system results, in particular, for stimulus frequencies exceeding 60 Hz.

The sensitivity variation highlights the importance of having a haptic system capable of independently and simultaneously providing multi-planar, variable intensity and frequency, mechanovibrotactile stimuli across the different areas of the finger. The results also demonstrate that the haptic system can provide stimuli 1) below and above the human sensitivity threshold and 2) with frequency within the sensitivity range of touch mechanoreceptors. The force applied by the handheld device (von Frey hairs) is different from the force applied by the haptic system. The contact area between the skin and the von Frey hair and the contact area between the skin and the actuated membrane of the fingertip interface can explain this difference. The diameter of the full contact area of the latter is 1.4 mm, which is significantly bigger

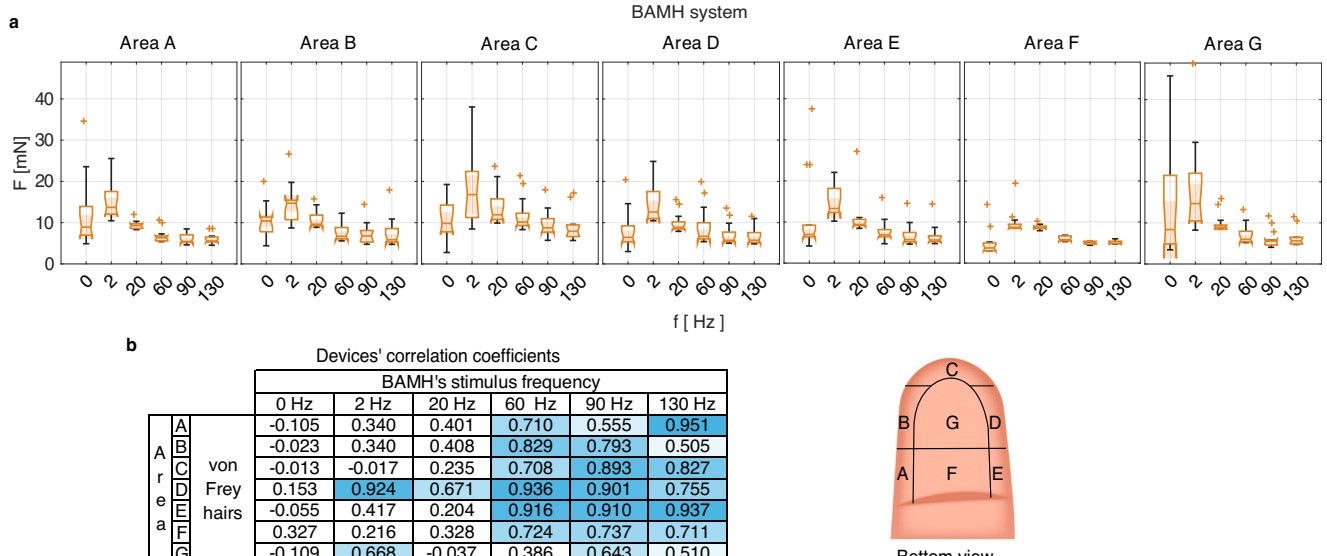

**Fig. 4 | Index distal phalanx sensitivity. a** Sensitivity across areas and BAMH's stimuli frequencies. The median is highlighted with a notch, + are outliers, and each box's limits are the second and third quartiles. **b** Linear correlation between the result obtained using the BAMH system and von Frey hairs (handheld device) across different areas of the fingertip. Light blue shading emphasizes coefficients exceeding 0.5, with increasing color intensity indicating proximity to a correlation coefficient of 1 (where 1 represents direct relationship of the variables).

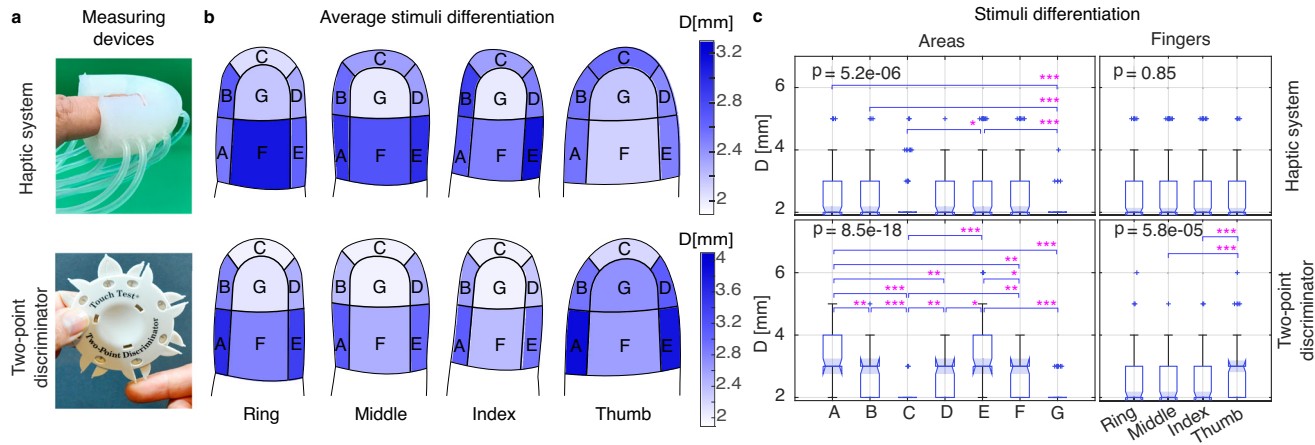

**Fig. 5 | Distal phalanx stimuli differentiation. a** Devices used. **b** Average stimuli differentiation minimum distance across fingers and areas, where the magnitude of the distance is reflected in the color intensity of the areas. **c** Stimuli differentiation minimum distance across fingers and areas, where the median is denoted with a notch, + are outliers, and each box's limits are the second and third quartiles. ∗, ∗∗, and ∗∗∗ represent statistical significance with $p \leq 0.05$, $p \leq 0.01$, and $p \leq 0.001$, respectively.

than that of the von Frey hairs (the largest diameter used is 0.18 mm). When using the haptic system, the direct interaction of the operator and participant is primarily limited to adjusting the fingertip interface. However, von Frey hairs are manually applied, so the operator's skills can affect the data collection. Factors such as the application angle between the von Frey hair and the skin, the hair buckling level, and the ability to stimulate the same spot can impact the data variability[33]. Another finding concerns the sensitivity results across frequencies resembling the Pacinian (FAII) response to various frequency stimulus[5]. This may be explained by results obtained by previous research[34], demonstrating that FAs mechanoreceptors are the most sensitive receptors in von Frey threshold determination.

## Distal phalanx stimuli differentiation

Human stimuli differentiation refers to the minimum distance participants need to establish whether one or two stimuli are applied. This measures the participant's tactile spatial resolution. The greater the minimum distance, the lower the tactile spatial resolution. For the haptic system, the distance refers to the edge-to-edge distance, $d$, of the paired chambers in an area. For the handheld two-point discriminator device (see Fig. 5a), distance is defined between the two rounded tips.

The fingertip interface results (see Fig. 5b, c varying color intensity and median values, respectively) establish that the tactile spatial resolution changes across areas (Kruskal Wallis, $n_A = n_D = 68$, $n_B = 69$, $n_C = n_G = 72$, $n_E = 71$, $n_F = 65$, $p = 5.18 \times 10^{-6}$), but insufficient evidence exists to state the variation across fingers (Kruskal Wallis, $n_{Ring} = 118$, $n_{Middle} = n_{Index} = 123$, $n_{Thumb} = 121$, $p = 0.85$). The pulp of the finger (area G) is among the areas with the higher tactile spatial resolution.

The handheld device results validate that stimuli differentiation changes across areas (Kruskal Wallis, $n = 40$, $p = 8.46 \times 10^{-18}$) and that the pulp of the finger is an area with one of the highest tactile spatial

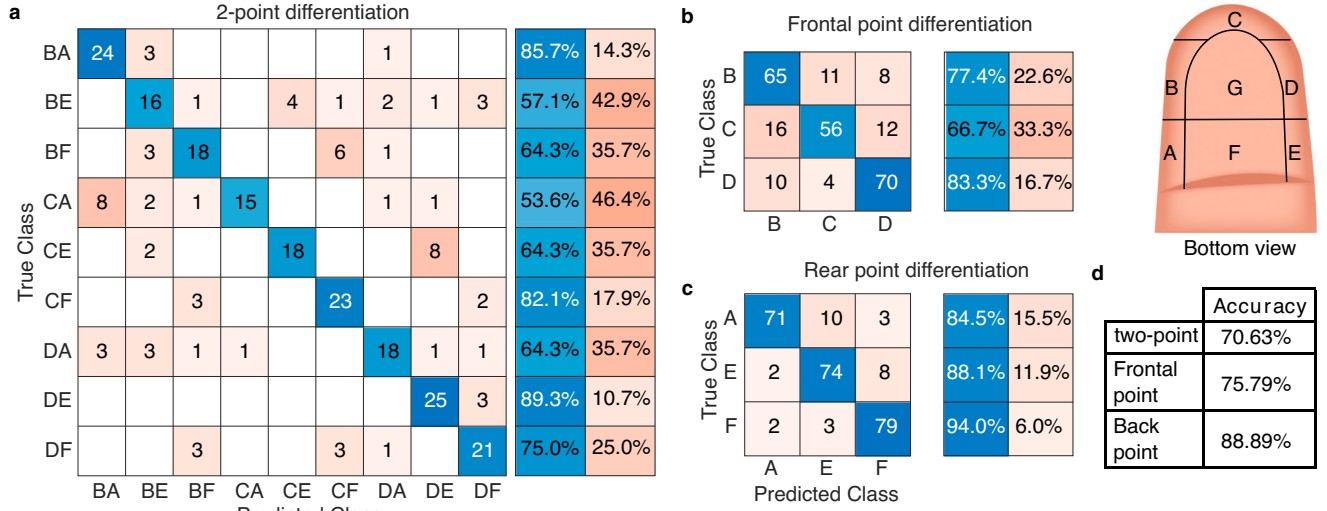

Fig. 6 | Index stimuli differentiation across areas. a Confusion matrix illustrating the ability of participants to recognize simultaneous stimuli applied in pairs in the frontal and rear area of the fingertip, where the pairs are denoted using the name of the areas, e.g., BA denotes that a simultaneous stimulus was applied in areas B and A. From these pair stimulus predictions, the confusion matrix for correctly identifying the stimulus in **b** the frontal areas and **c** the back areas was obtained. **d** Participants' prediction accuracy.

resolutions (see Fig. 5b, c). Results also vary across fingers (Kruskal Wallis, $n = 70$, $p = 5.77 \times 10^{-5}$). More specifically, the thumb's tactile spatial resolution differs from that of the rest of the studied fingers (Wilcoxon rank sum test, $n_{\text{Thumb}} = 70$, $n_{\text{Other fingers}} = 210$, $p = 1.49 \times 10^{-5}$).

The stimuli differentiation change across areas confirms the need for a haptic system capable of providing multiplanar (frontal, bottom, and lateral fingers' areas) and independent variable intensity stimuli to the distal phalanx. Several factors could explain the different results across devices for fingers. The handheld device is used to apply two stimuli simultaneously. Simultaneity and repeatability of force intensity and stimuli location cannot yet be guaranteed across trials[35]. The data collection with the haptic system is less operator-dependent. Additionally, stimuli can be applied simultaneously with the same intensity at the same spot across trials. The discrepancy regarding the change of stimuli differentiation across fingers between the BAMH results and those from the two-point discriminator can be explained by the difference in height, length, and width between the thumb and the rest of the fingers (Kruskal Wallis, $n_{\text{Thumb}} = 18$, $n_{\text{Other fingers}} = 54$, $p_{h_f} = 5.16 \times 10^{-4}$, $p_{l_f} = 4.71 \times 10^{-8}$, and $p_{w_f} = 3.4 \times 10^{-9}$). Consequently, it is necessary to use a modified fingertip interface to enhance the evaluation of the thumb's stimuli differentiation capabilities.

## Stimuli classification across different stimuli distances

To investigate human touch further, we used the data collected for the stimuli differentiation to analyse the participants' stimuli classification performance using four haptic interfaces. The criteria employed include accuracy (the fraction of samples that the participants correctly classified), precision (the fraction of samples classified as two-point stimuli that were actually two-point stimuli), classification sensitivity (the fraction of all two-point stimuli samples that were correctly classified), and false positive rate (the probability of false alarm). Their equations are presented in Methods.

The overall accuracy and precision (presented in Supplementary Fig. 6) demonstrate that both change across areas, fingers, and the distance between stimuli. Consistent with the sensitivity and differentiation results, the pulp of the finger (area G) is among the areas with the highest accuracy and precision across the four distances. The frontal area C also has high values. The green ellipses in Supplementary Fig. 6 highlight that having a bigger distance does not guarantee high

accuracy and precision. Thus, low accuracy and precision were observed at the bottom of (area F) the ring and middle fingers.

The participant's classification sensitivity and false positive rate are illustrated in Supplementary Fig. 7. In some areas, the participants' best performance was with the fingertip interface with a 4 mm stimuli distance instead of the 5 mm distance (see fuchsia rectangle in Supplementary Fig. 7). Yellow rectangles highlight areas where the performance across distances is similar. The green rectangle shows the closest result to the diagonal; this implies that the performance for a 2 mm stimuli distance for this area and finger was the closest to random guessing.

The high accuracy and precision observed in areas C and G (finger's front and pulp) explain why they are used for braille reading[36] and palpation[37]. The low accuracy and precision findings can be explained by the fact that although the system provided stimuli for all the trials, some participants did not feel it (see Supplementary Fig. 8). The fingers and areas with the highest number of trials where the participant did not feel a stimulus correspond to those with low accuracy, precision, and classification performance. This outcome could be due to sex because of females' lower median values of finger height for the ring and middle finger (see Supplementary Fig. 9). This influence of sex will be further evaluated as part of future work. Additionally, the negative distance between the tip of the nail and the tip of the finger, $d_{\text{nf}}$, demonstrates that the participants had short nails. So, long nails did not affect the results.

## Index stimuli differentiation across areas

The results presented in Fig. 6 reveal variation in human stimuli differentiation accuracy despite using identical vibrotactile pulse stimuli in both regions. Notably, combinations characterized by areas' alignment along the finger (DE, BA, and CF) exhibited the highest percentage of correct pair identification across trials, with coefficients of 89.3%, 85.7%, and 82.1%, respectively (see Fig. 6a). Furthermore, participants mainly confused the CA pair with the BA pair and the DE pair with the CE pair. This can be explained by the fact that participants commonly misidentify stimulation in areas B and D as stimulation in area C (see Fig. 6b). It is also worth mentioning that stimuli differentiation in rear areas (ranging from 84.5% in area A to 94% in area F, see Fig. 6c) surpasses that in frontal areas

(ranging from 66.7% in area C to 83.3% in area D). This discrepancy may be attributed to the variability observed in index finger sensitivity results (see Fig. 4a), where sensitivity to 20 Hz stimuli exhibits low variability in area F, while area C demonstrates high variability. This is further corroborated by the accuracy findings presented in Fig. 6d. In particular, the participants' prediction accuracy for correctly identifying the pairs is 70.63%. From these results, it was found that participants' accuracy in identifying the rear area of the pair is 88.89%, whereas that for frontal areas is 75.79%.

The findings from this research are significant in three aspects. First, this research proposes a haptic system that, through a soft haptic interface, provides simultaneous mechano-vibrotactile, multiplanar, and variable intensity stimuli within the skin's mechanoreceptors' sensitivity bandwidth (from 0 Hz to 130 Hz). Therefore, enables better selectivity on the touch mechanoreceptor's activation across the front, bottom, and lateral areas of the finger. Second, with the BAMH system's range, modularity, and portability, the system can be used in several fields, e.g., neuroscience, psychology, and sociology, to further study the sense of human touch from a broader perspective. Third, this system quantifies sensitivity and stimuli differentiation of the distal phalanx in human fingers while reducing operator bias.

The haptic system could address the needs of clinicians[38] for operator-independent, accurate data of sensitivity and stimuli differentiation. In addition, the haptic system does not require specialized training. The operator's role will be to put the haptic interface on the fingers of the patient and start the test using the computer-machine interface; data acquisition can be made automatically. Furthermore, the proposed device has the potential to be used as a diagnostic device to assess and monitor the loss of touch, e.g., carpal tunnel syndrome or diabetic neuropathies. Its resolution makes it suitable for gathering healthy participants and patient datasets that could be later used to train Artificial Intelligence algorithms for sense-of-touch deterioration diagnostics. Taking into account the evaluation frameworks[39,40] of haptic systems, our device's versatility can enhance the touch feedback performance of teleoperated systems, virtual communication, and virtual reality and augmented reality systems. These enhancements can impact various fields, including socialisation, healthcare, and education. For instance, the fingertip interface could be embedded into gloves to assist social and behavioural scientists with more realistic experiences. Using the system in virtual social interaction applications will help understand touch's role in social bonding.

Future studies are required to obtain insights and further understanding in the area of the human sense of touch. For instance, by working with neuroscientists and social scientists, we can use our BAMH system to investigate individually the components related to this sense and the individual and combined contribution of each mechanoreceptor. We envision studies on how fingertip interface personalisation (e.g., fitted to the user fingers' dimensions), sex, environment/social context, and stimulus' location, type, intensity and application time affect the stimuli perception. This understanding is important to support the development of haptic feedback systems capable of providing more realistic feedback.

Future work surrounding our haptic device includes broadening the frequency range of vibrotactile pulse feedback, aiming to reach 400 Hz (thereby expanding the stimulation range for FA II mechanoreceptors in our system). The spatial resolution of the BAMH system could be maximised to match that of existing systems such as those described by refs. 15,16. This exploration aims to improve touch simulation fidelity while maintaining the versatility and intrinsic safety of our BAMH system. Enhancements in this area will widen the fields of application, for instance, to those requiring lower intensity interactions, e.g., during manipulation activities (in Virtual Reality environments) as well as drilling and those combining haptic and braille technologies[41]. Additionally, understanding individual mechanoreceptor functions is crucial, as is integrating additional feedback modalities like temperature, social context, and shear forces to enhance user immersion and experience.

## Methods

All participants recruited for our studies were volunteers. They gave informed consent for inclusion before the studies. The studies were approved by the University Research Ethics Committee (ID: 17503/001).

### Fingertip interface and flat sample manufacturing

The soft fingertip interface and the flat samples were cast from Dragon Skin™ 20 Smooth On mixed in a ratio of 1:1 from parts A and B. Supplementary Fig. 1a shows the 3D printed mould parts used to fabricate the fingertip interface. To improve the adaptability of the haptic interface to the finger curvature, we used a human index finger mould for the shape; the mould measures 18 mm in width and 27 mm in length. Two pillars were used per area to create the two chambers after moulding. The pillar's diameter of 1.4 mm and height of 5 mm formed a chamber, and the 2.2 mm diameter base formed the pipe placement channel (see Fig. 1b, c). Once assembled (see Supplementary Fig. 1b), the distance between the finger mould and the top of the pillar is 0.5 mm. After pouring Dragon Skin™ 20, this gap becomes the 0.5 mm thick soft membrane (see Fig. 1c and Supplementary Fig. 1c). This membrane presses into the participant's finger to stimulate it. The membrane is actuated by applying pressurized air through the chambers. Each chamber is airtight connected to a silicone pipe (1.5 mm inner diameter (ID)) through the pipe placement channel. This silicone pipe is glued to a 4 mm outer diameter (OD) pneumatic pipe to connect the chamber to the output of the valves (see Fig. 2a–c). To improve the adaptability of the fingertip to wide fingers, two incisions were made in the frontal part of the fingertip (see Supplementary Fig. 1e). These incisions also allowed visual confirmation that the frontal part of the finger was touching area C, the fingertip interface frontal area. The thickness of the silicone sleeve is 9 mm. Similarly, Supplementary Fig. 10a–c illustrates the 3D printed parts used in the fabrication of the flat samples of the fingertip interface used for the characterisation of the haptic system. The fabrication process of this sample follows the same procedure employed for fabricating the fingertip interface.

### Haptic feedback system components description

The haptic system (Fig. 2) comprises the bioinspired soft haptic fingertip interface and the control.

After manufacturing the fingertip interface, the following components can be observed: the sleeve, the internal chambers, and the Hook-and-Loop fastener bands that tighten the haptic interface to the finger (see Fig. 1b, c and Supplementary Fig. 1). The membrane (1.4 mm ID and 0.5 mm thickness) and the electro-pneumatic control subsystem allow the participant's finger to be stimulated.

The electro-pneumatic control subsystem (see Fig. 2a–c) is composed of a printed circuit board - PCB (that contains the control, sensing, and actuator conditioning circuits), the fast switching valves (highlighted with yellow rectangles), the proportional pressure regulator (highlighted with green rectangles), and the electric and pneumatic power supplies. The Teensy® 4.1 board contains the microcontroller of the system. The microcontroller has 22 independent timers to simultaneously provide up to 22 independent frequency PWM outputs, 18 analogue inputs with a 10-bit Analog to Digital Converter (ADC) resolution, and USB serial ports to communicate with external devices such as a computer. One PWM output (with a duty cycle $\delta$ from 0% to 100%, which is defined using a variable from 0 to 255, respectively), together with a second-order Butterworth filter, permits modulating the internal pressure of the system through the proportional pressure regulator (Camozzi K8P, output range: 0 kPa to 300 kPa). Fifteen PWM outputs are employed to control the 15 fast

switching valves (FESTO, MHE3, 3/2 ways, normally closed, -0.9 kPa to 800 kPa, 280 Hz maximum switching frequency) using the valve actuation circuit (highlighted with blue rectangles). Fourteen of these valves are connected to the 14 chambers of the fingertip interface. We refer to the fifteenth valve as a dummy valve because its output is disconnected/open. This valve is actuated during the stimulus differentiation tests when only one chamber is actuated. Actuating the dummy valve prevents participants from using audio feedback to differentiate one point from the two-point stimulus. Two Honeywell heavy-duty pressure sensors (PX2, absolute pressure, 50 psi range, 5 V; highlighted with magenta rectangles) connected to the ADC inputs of two microcontrollers acquire the pressure of the system (measured at the output of the pressure regulator), and the internal pressure of a chamber (measured at the output of one switching valve). The sampling frequency is set to 1 kHz. A compressor with a reservoir and electrical power supply provided the power required by the electropneumatic system. For safety during the experiments with participants, the maximum value of the general internal pressure of the system was set to 150 kPa.

### BAMH system characterisation

The experimental setup of the flat sample is illustrated in Supplementary Fig. 10d. The chamber's internal pressure versus the force applied by the membrane was obtained using a Tedea Huntleigh Model 1004 Single Point Load Cell with a range of 3 N. The load cell was mounted on a motorized linear rail (Zaber X-LSM100A) opposite a chamber. Forces from the load cell were acquired via a National Instruments Data Acquisition System (DAQ). The load cell was moved into contact with the chamber with a 200 mN pre-load. The chamber was then preconditioned by loading and unloading it three times with 178.2 kPa.

The steady/mechanotactile stimulus provided by the haptic system was evaluated using steady-state tests. This denotes that the actuation signal of the valves had a duty cycle of 100%. Training data was obtained by increasing the input pressure from 10.6 kPa to 178.2 kPa and back three times in steps of 1.78 kPa. The sensor was re-zeroed between each cycle. Testing data was obtained by stepping the input pressure from 15.8 kPa to 178.2 kPa in steps of 4.39 kPa, returning to 0 kPa between each step; this was repeated four times. All pressure steps had a duration of 1.5 seconds. During steady-state tests, data from the load cell was sampled at 2.5 kHz and fed through a 100 Hz low-pass filter.

The vibrotactile pulse stimulus provided by the haptic system was evaluated using modulated tests. The tests were conducted using the valves' actuation signal frequency of 20 Hz and duty cycle of 5%. Training and testing data were obtained at the same pressures as steady-state tests. Modulated tests were also conducted to find the applied force corresponding to a range of frequencies and duty cycles. In line with the index finger stimuli sensitivity across several frequencies test, vibrotactile pulse data was obtained using $\delta = 75\%$, frequencies of 2 Hz, 20 Hz, 60 Hz, 90 Hz, and 130 Hz, and a pipe length of 250 mm. This data was filtered and peaks were extracted in line with the modulated test methods. Data from the load cell was sampled at 5 kHz and fed through a 200 Hz low pass filter, then force peaks were extracted over each period (1/frequency). Additionally, in line with the differentiation test, vibrotactile pulse data was obtained at a set system output pressure of 154.8 kPa. Then, the frequency was increased from 20 Hz to 180 Hz in steps of 5 Hz at duty cycles of 5%, 10%, 25%, 50%, and 75%; this was repeated twice. These tests were repeated with pipe lengths of 60 mm, 150 mm, and 250 mm and repeated after one and two weeks using the same chamber and a different chamber from a second flat sample, referred to as flat sample 2.

Power analysis was conducted to identify that the minimum sample size for evaluating the haptic feedback system was three. For the calculation, we used handheld devices' experiment data ($\sigma_{sensitivity} = 1.1$ mN and $\sigma_{stimuli\ differentiation} = 0.83$ mm), a power of 0.8 and a significance level of 0.01.

### Sense of touch evaluation using the fingertip interface

Participants for the evaluation of stimulus sensitivity and stimuli differentiation across fingers and areas were 11 males and 7 females between 21 and 39 years old. On the other hand, 11 males and 3 females participated in evaluating the index finger sensitivity across frequencies and its stimuli differentiation across areas. The participants did not have any sensory or motor impairment. The evaluated fingers were the right hand's thumb, index, middle, and ring fingers. The test was performed in the seven areas illustrated in Fig. 1d. The stimulus/stimuli were applied for 1.5 seconds (0.5 seconds less than the stimulation time used in ref. 42) to limit the length of the experiment to one hour. The participants also used ear defenders to decrease the audio feedback due to the actuation of the valves and pressure regulator. At the end of the test, each participant completed a questionnaire regarding sex (biological attribute), age, and profession. During the experiments, the stimulus intensity was changed by setting the output pressure of the system between 0 and 154.8 kPa. This maximum pressure was selected to minimize the risk of accidental damage to the fingertip interface. This maximum pressure is achieved by providing a steady/mechanotactile stimulus and setting the pressure regulator's PWM duty cycle variable to 170. Where a value of 255 in this variable implies 100% duty cycle of the pressure regulator's PWM, approximately 300 kPa.

**Sensitivity analysis/one-point differentiation test.** This test determines the chamber's minimal force/internal pressure required for the participant to feel a stimulus across fingers and areas. Firstly, the participant was asked to rest their arm comfortably (see Fig. 2c). The fingertip interface, with the hook-and-loop fastener straps, was tightened to the finger with 2 mm edge-to-edge chamber pair internal distance. Using the fingertip interface, a steady stimulus was applied by actuating one chamber with a duty cycle $\delta = 100\%$ for 1.5 seconds (Supplementary Fig. 2a illustrates the raw signal). The system then waited one second before applying a new stimulus. Participants were stimulated with incremental pressure. This was achieved by increasing the duty cycle of the pressure regulator's PWM signal using the duty cycle variable in the microcontroller, which ranges from 0 to 170. The latter corresponds to having the defined maximum internal pressure of the chambers at approximately 150 kPa. This variable, initialized at 0, was initially incremented in steps of 8. Once the participant feels the stimuli, the system decreases the current value of the duty cycle variable by 12. Then, for greater accuracy, the system repeats the process of incrementing the duty cycle variable in steps of one unit until the participant feels the stimulus again. This process is performed in the seven areas of each finger, one area at a time. The areas were selected randomly, and one sample per area per finger was taken for each participant. The data collected included the pressure regulator's PWM variable, the internal pressure of the chamber, and the examined area and finger.

**Index finger sensitivity analysis test across frequencies.** This test determines the chamber's minimal force required for the participant to feel a vibrotactile pulse stimulus with $\delta = 75\%$. The evaluated frequencies were 2 Hz, 20 Hz, 60 Hz, 90 Hz, and 130 Hz. These frequencies were chosen considering the minimum overlap between the sensitivity frequency range of the mechanoreceptors. The Wilcoxon rank sum test with a 1% significance level was used to determine the frequency at which human participants required the highest stimulus intensity to recognise the stimulus. The rest of the methodology was repeated in line with the sensitivity analysis across the fingers previously described.

**Stimuli differentiation analysis within areas.** The two-point differentiation test defines the minimum distance humans need to establish whether one or two stimuli are being applied within one area. This test was performed across all seven areas and the four fingers. For this test, fingertip interfaces with different distances (edge-to-edge distance of 2 mm, 3 mm, 4 mm, and 5 mm) between the paired chambers in one area were used. This distance was kept constant across all areas in a fingertip. Previous studies[43,44] showed that using low-frequency vibrations with a low-duty cycle leads to better spatial differentiation in humans with lower power consumption (there is no statistical difference when using 50% duty cycle than when using low-duty cycle[43]). Therefore, we chose a PWM signal of 20 Hz with a duty cycle of 5% for the valve actuation (Supplementary Fig. 2b illustrates the raw signal). The stimulus intensity was defined by setting the pressure of the system to 154.8 kPa. These values were kept constant during all differentiation tests. Ten trials were performed per area, with five trials for activating both chambers simultaneously and five trials for activating only one chamber. The order of application of these stimuli was randomized. To avoid the participants using the audio feedback from the valves when providing one stimulus, the dummy valve and the corresponding area valve were activated. The data collected using the haptic system was the number of stimuli felt by the participant (zero, one, or two), the actual number of actuated chambers, and the area and finger examined. The minimum distance that the participant needed to differentiate the stimulus was set by the smallest distance where the participant answers were ≥60% correct. This means the participant answers were correct for at least 6 out of 10 trials.

**Index stimuli differentiation analysis across areas.** This test evaluates the participants' abilities to identify pairs of stimuli, one at the rear of the index finger (areas A, F, and E) and one at the front (areas B, C, and D). These combinations are similar to those used in[45] in their assessment of multi-contact actuation. The stimuli comprised a single point in both areas, actuated simultaneously, at a pressure of 154.8 kPa, $f = 20$ Hz, $\delta = 5\%$ for 1.5 s. The participants were presented with a sheet showing top views of stimuli pair locations numbered from 1 to 9 (see Supplementary Fig. 11). For training, each stimulus was displayed twice to each participant. Then, the stimuli pairs were presented in a random order, with each stimuli pair being presented twice to give a total of 18 pairs. After each pair of stimuli were presented, the participant was required to identify the stimuli pair number using the sheet.

**Human sense of touch evaluation using handheld devices**
The participants were 5 males and 5 females between 20 and 37 years old. The participants did not have any sensory or motor impairment. Similar to the fingertip evaluation methodology, the evaluated fingers were the thumb, index, middle, and ring fingers of the right hand. The test was performed in the seven areas illustrated in Fig. 1d. At the beginning of the experiments, the participants completed a questionnaire regarding their sex, age, and profession.

**Sensitivity analysis.** Participants were blindfolded. Following a similar procedure as described for the fingertip interface, they were also asked to rest their right arm in a comfortable position. The finger was touched at one location with an incremental force until the participant was able to feel the stimulus. We used the five von Frey hairs with the lowest force magnitude from Aesthesio®to apply incremental force (illustrated in Fig. 3a). This procedure was repeated across all seven areas of the four fingers.

**Stimuli differentiation analysis.** We used the two-point orientation differentiation test. Participants were blindfolded and asked to rest their right arm comfortably. The four fingers were stimulated seven

times in each area with two points (four times horizontally/across the finger and three times vertically/along the finger) and twice with one point. The order of application was randomized. Participants indicated whether one or two points were applied. If the participant's answer was two points, they also indicated whether the stimuli were applied horizontally or vertically. The stimuli were applied using a two-point discriminator (Fig. 3b) with an applied pressure around the point of blanching. The minimum distance that the participant needed to differentiate the stimulus was defined by the smallest distance where the participant correctly identified both one-point stimuli and the two-point answers were ≥60% correct. This means at least four correct answers out of seven trials for the two points.

**Data analysis**
The haptic system characterisation was performed using Matlab R2021b. The data was split at every point where the input pressure or frequency was changed. To remove the effects of these transitions, data was removed for 0.15 seconds before and 0.5 seconds after each transition. All data was zeroed to its preceding 0 kPa step. Training data for $\delta = 100\%$ across all tests were used to establish the relationship between the pressure regulator input and the pressure at the output of the valve. This was done by fitting this data to a 4th-order polynomial. For all tests where the duty cycle was less than 100%, force peaks were extracted as the maximum force over each period $1/f$. The force-pressure relationship was determined for each test by fitting training data to a 2nd-order polynomial. This fit was then compared to the testing data to find the root mean square deviation (RMSD). Hysteresis was extracted from cubic splines of training data during pressure loading and unloading, whilst non-repeatability was extracted from testing data. These were both done in line with BS EN 61298-2:2008[46]. For modulated tests, average peak forces were extracted for every duty cycle, frequency, and pipe length combination in each test. The combinations were considered usable if they produced an average peak force larger than the maximum peak force for 0 kPa and lower than 70 mN, as forces above this suggested the valve was saturating. The highest usable frequency across three of four tests was extracted as the maximum frequency for each duty cycle and pipe length combination. The correlation analysis between duty cycle, pipe length, and max frequency was done in Microsoft Excel Version 2307 using the Regression tool in the Data Analysis toolbox.

The frequency analysis of the force utilises the Fast Fourier Transform (FFT) of the force-time series data. The FFT was obtained using the FFT function in Matlab, where the DC component was removed by mean subtraction, consistent with the frequency analysis methods described in ref. 47. The Fast Fourier Transform (FFT) accuracy is affected by the time series length, which should be $2^n$, where $n \in \mathbb{R}$, and the time series must contain a whole number of periods of the pulse signal. The raw data contains noise, so these requirements are not satisfied to analyse the magnitude of the spectral coefficient. Consequently, this analysis focuses on the capability to deliver the pulse signal at the desired frequency.

The remaining statistical analysis was performed using Matlab R2021b. We employed the non-parametric Kruskal-Wallis test with the Bonferroni correction to evaluate if the results from the sensitivity and stimuli differentiation tests were statistically different across fingers and areas. The Wilcoxon rank sum test with a 1% significance level was used to determine if the pressure, force, or minimum distance was higher or lower across fingers and areas.

We used a confusion matrix to analyse the human classification performance across 2 mm, 3 mm, 4 mm, and 5 mm stimuli distance and across the index areas. This matrix was calculated using the confusionmat function from Matlab R2022a. This matrix was also used to calculate the accuracy, precision, sensitivity, and false positive rate

using the following equations:

$$\text{Accuracy} = \frac{TP_1 + TP_2}{TP_2 + TN_2 + FP_2 + FN_2} \tag{1}$$

$$\text{Precision} = \frac{TP_2}{TP_2 + FP_2} \tag{2}$$

$$\text{Sensitivity} = \frac{TP_2}{TP_2 + FN_2} \tag{3}$$

$$\text{False positive rate} = 1 - \frac{TN_2}{TN_2 + FP_2} \tag{4}$$

where $TP_1$ and $TP_2$ are the numbers of samples where participants correctly identified the one-point and two-point stimuli, respectively. $TN_2$ is the number of samples where participants correctly classified the stimulus as not two-point stimuli. $FP_2$ and $FN_2$ refer to the number of samples where participants incorrectly classified as two-point and not two-point stimuli, respectively. Participants' classification sensitivity and false positive rate were also used in the Receiver Operator Curve (ROC) space. This allowed analysing the two-point stimuli classification performance of the participants using the fingertip interface.

### Reporting summary

Further information on research design is available in the Nature Portfolio Reporting Summary linked to this article.

## Data availability

The data generated in this study have been deposited in the University College London Data Repository under https://doi.org/10.5522/04/26169838[48].

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

## Acknowledgements

This work was supported by the U.K. Engineering and Physical Sciences Research Council (EPSRC) under Grants No. EP/SO14039/1 and EP/V01062X/1 (H.W.), the EPSRC-IAA-UCL Therapeutic Acceleration Support (TAS) Fund - Call 10 under Grant No. 184646 (S.-A.A., H.W., and M.K.) and the U.K. Royal Society Research Grants - 2022 Round 2 under Grant No. RGS\R2\222228 (N.H.).

## Author contributions

S.-A.A and H.W. conceived the system and designed the research; S.-A.A. designed the experiments with support from H.W. and N.H.; D.R. performed the characterisation of the haptic system using the flat sample; he also designed and performed the stimuli differentiation across areas experiments with the support of S.-A.A., H.W., N.H., and M.K.; S.-A.A. prepared the figures and performed the sensitivity, stimuli differentiation experiments, and frequency analysis with the support of H.W. and N. H.; and S.-A.A. wrote the manuscript. All authors contributed to the data analysis and discussions and reviewed the manuscript.

## Competing interests

The authors declare no competing interests.
