## [Peer Review File · Nature Communications]

Bioinspired adaptable multiplanar mechano-vibrotactile haptic systemREVIEWER COMMENTS

Reviewer #1 (Remarks to the Author):

The paper presents the development and assessment of a new haptic system for fingertip mechano-vibrotactile stimuli. The proposed system is interesting in my opinion and presents good potential. The evaluation procedure is overall accurate, and the results are rigorous and properly elaborated.

Validity

The paper introduces a new haptic system for accurately rendering mechanical and vibration stimuli to the fingertip. The focus of the paper is on the proposed display and its potential to investigate tactile perception.

Significance

The investigation of the sense of touch is currently an important research and technology challenge with potential applications in several human/human and human/machine contexts. The availability of systems such as the one proposed in the paper is important to support the research in this field. The proposed device is developed and assessed in a rigorous way. In my opinion, it could be better framed with respect to the existing solutions.

Data and methodology

The approach for the assessment of the device is formally correct and well explained. Data are collected and elaborated properly, results are clearly presented and interpreted. However, I found some limits in the presented results, I pointed out them in the following.

Analytical approach

The approach is rigorous, and the statistical analysis of data is correct and well presented in my opinion.

Some points that could be improved in the paper

In the introduction, the author(s) briefly summarize(s) the state of the art in haptic device development and propose the presented device as a solution for the existing gaps in current technologies. However, it is not clearly explained which are the specific gaps, which of them have been fulfilled, and which of them are still open.

The motivation for some design choices could be provided, for instance: why pneumatic actuation was employed? How was the material for the interface chosen?

In the assessment section, a series of data regarding sensitivity, differentiation, and classification are presented and discussed. It is not clear in my opinion if the aim of this part of the paper is to assess the capabilities of the proposed device or to evaluate such information about tactile perception in humans. According to the introduction of the paper, the aim of the experiments should be the assessment of the device. Still, in this case, the obtained results should be compared with some reference data or other devices.

Some results are presented in a rather synthetic way: for instance, a differentiation among genders is identified and discussed, but I think that this difference could be motivated by different aspects that probably are out of the scope of the paper, and would require deeper investigations and more subjects. I would therefore avoid the discussion on the influence of the subject gender in this context.

Reference

The overall review of the state of the art is proper in my opinion. I liked the initial introduction to the sense of touch, which is clear and synthetic. As I highlighted in the previous point, I suggest clarifying which gaps and limits of the state-of-the-art have been solved by the developed device and which ones are still open.

Reviewer #2 (Remarks to the Author):

Authors developed a pneumatic multiplanar display that can present stimuli up to 299.2mN in intensity and 130Hz vibration. The authors evaluated the absolute and discrimination threshold of the force.

The strength of this study is to develop a system that can provide spatially distributed vibration up to 130Hz.

I have the following concerns:

1.

While the Pacinian-FAII's most sensitive frequency is over 200Hz [1], the developed display cannot present such frequencies. In our daily lives, we recognize objects' haptic properties and perform haptic interactions with objects using vibrotactile information containing over 200Hz [2]. If researchers aim to develop a display for application scenarios like those mentioned by the authors (such as training and VR, etc.) that aim to simulate everyday haptic interactions, the display should naturally need to support this frequency range.

[1] Jones, L. A., & Lederman, S. J. (2006). Human hand function. Oxford university press.

[2] Shao, Y., Hayward, V., & Visell, Y. (2016). Spatial patterns of cutaneous vibration during whole-hand haptic interactions. *Proceedings of the National Academy of Sciences*, 113(15), 4188-4193.

2.

The haptic displays that performed multiplanar actuation have been developed [3][4]. This is inconsistent with the authors' note "these current haptic tactile solutions are mainly focused on stimulating the pulp of the fingers and have limited or no simultaneous multiplanar actuation." I have another concern about the sparse spatial resolution of the author's display. The spatial resolution is important in this type of finger-mounted display because the spatial threshold of the fingertip is about 2-3mm.

[3] Taniguchi, T., Sakurai, S., Nojima, T., & Hirota, K. (2018). Multi-point pressure sensation display using pneumatic actuators. In *Haptics: Science, Technology, and Applications: 11th International Conference, EuroHaptics 2018, Pisa, Italy, June 13-16, 2018, Proceedings, Part II* 11 (pp. 58-67). Springer International Publishing.

[4] Ujitoko, Y., Taniguchi, T., Sakurai, S., & Hirota, K. (2020). Development of finger-mounted high-density pin-array haptic display. *IEEE Access*, 8, 145107-145114.

3.

The purpose of the perceptual experiments is not clearly stated. Initially, it seemed like the aim was to compare the thresholds of real stimuli (von Frey hairs) and those produced by the display to demonstrate a similarity in trends. However, if that were the case, it would be better to examine the correlation between the thresholds of both. Why were gender, area, and threshold variations across different fingers compared, and how did this lead to evaluating haptic display?

4.

It's essential to provide information about the vibration waveforms presented to the participant's fingers. While the authors mention driving with PWM signals, the actual waveform presented is unlikely to be a perfect square wave. Perception can be influenced by the shape of the waveform applied to the human skin.

5.

In the experiments involving human subjects, the authors restricted the PWM signal frequency to 20Hz. Why was this specific frequency chosen? A study limited to 20Hz would primarily activate FAI, but does this mean FAII can be disregarded?

Minor:

- The second paragraph in the Introduction seems overly detailed. It would be more concise to focus on aspects directly relevant to the study.

A bioinspired adaptable multiplanar mechano-vibrotactile haptic system (NCOMMS-23-44661-T): Reviewers comments and answers

DBPR

The authors express their gratitude to the Reviewers for dedicating their time to reviewing our work and providing us with insightful comments. We have addressed point-by-point all comments and concerns raised, which we believe has substantially improved the manuscript. Additionally, a version of the paper delineating the alterations made throughout is appended to this response.

Reviewer 1

Comment 1

The paper presents the development and assessment of a new haptic system for fingertip mechano-vibrotactile stimuli. The proposed system is interesting in my opinion and presents good potential. The evaluation procedure is overall accurate, and the results are rigorous and properly elaborated.

Validity:

The paper introduces a new haptic system for accurately rendering mechanical and vibration stimuli to the fingertip. The focus of the paper is on the proposed display and its potential to investigate tactile perception.

Significance:

The investigation of the sense of touch is currently an important research and technology challenge with potential applications in several human/human and human/machine contexts. The availability of systems such as the one proposed in the paper is important to support the research in this field. The proposed device is developed and assessed in a rigorous way. In my opinion, it could be better framed with respect to the existing solutions.

Data and methodology:

The approach for the assessment of the device is formally correct and well explained. Data are collected and elaborated properly, results are clearly presented and interpreted. However, I found some limits in the presented results, I pointed out them in the following.

Analytical approach:

The approach is rigorous, and the statistical analysis of data is correct and well presented in my opinion.

Response: We thank the reviewer for the time dedicated to reviewing this article and for

providing constructive comments that we now addressed. We revised and strengthened the manuscript accordingly.

Comment 2

In the introduction, the author(s) briefly summarize(s) the state of the art in haptic device development and propose the presented device as a solution for the existing gaps in current technologies. However, it is not clearly explained which are the specific gaps, which of them have been fulfilled, and which of them are still open.

Response: Many thanks for the comment. In the area of haptic device development, a number of gaps persist due to the multifaceted nature of the sense of touch, which is influenced by various factors such as social context, temperature, shear force exertion, and characteristics of tactile stimuli [1, 2, 3, 4]. Existing gaps [5, 6] include finding solutions to questions about how to achieve enhanced levels of realism and fidelity in haptic devices, how to integrate multimodal sensations, spanning steady mechanotactile, vibrotactile feedback, or temperature, how to miniaturise both the overall interface and its actuators to minimise user distraction and optimise focus solely on the tactile experience [5, 6]. Overcoming these hurdles is imperative for advancing haptic technologies and unlocking their full potential in various applications.

Our work, in particular, focuses on enhancing the fidelity of touch feedback, providing a range of haptic feedback sensations, and allowing wear-ability as our Bioinspired Adaptable Multiplanar Haptic (BAMH) system device:

1. is able to provide both mechanotactile/steady as well as vibrotactile stimuli with variable intensity over a wide range of frequencies. Hence, the device can stimulate mechanoreceptors including the SAII (sensitive to steady stimulus), the SAI (sensitive to vibrotactile stimulus with a frequency lower than 5 Hz), the FAI (sensitive to vibrotactile stimulus with a frequency between 5 Hz and 50 Hz), and the FAII (sensitive to vibrotactile stimulus with a frequency between 40 Hz and 400 Hz).
2. is capable of providing simultaneous stimuli on several planar areas of the entire fingertip surface, i.e., the frontal, lateral, and bottom areas of the finger.
3. allows a level of personalised design, adapting to the fingers' curvature. Haptic feedback is given through pressurised air actuation from a control subsystem to the tethered soft fingertip interface.

A number of existing challenges remain with regard to our developed BAMH haptic device. Foremost among these is the investigation how to broaden the frequency range of the vibrotactile feedback, our device is able to provide. Our ambition is to push its vibrotactile stimulus frequency spectrum towards 400 Hz. This expansion promises enhanced fidelity and realism in touch simulations. Furthermore, a deeper understanding of the distinctive functions of individual mechanoreceptors related to the sense of touch remains. Integrating additional haptic feedback modalities, such as temperature, social context, and shear forces, into our system represents another critical gap. Achieving this integration would enrich the user experience and provide greater immersion. For clarity, we moved the results from the characterisation of the BAMH system from the Methods Section to the beginning of the Results Section. Additionally, we now updated the Introduction and Future Work as follows:

Sec. **Introduction**, p. 1: "However, a number of gaps persist in the area of haptic device

development due to the multifaceted nature of the sense of touch, influenced by various factors such as social context, temperature, shear force exertion, and characteristics of tactile stimuli [1, 2, 3, 4]. Existing gaps include [5, 6] challenges enhancing levels of realism and fidelity in haptic devices, integrating multimodal sensations effectively, encompassing mechanotactile/steady and vibrotactile feedback, or temperature variations, and miniaturising both the overall interface and its actuators to minimise user distraction and optimise focus on the tactile experience [5, 6]. Finding solutions to these challenges is essential for advancing haptic technologies and unlocking their full potential across various applications.

Here, we show that our work contributes to enhancing the fidelity of touch feedback, providing a frequency range of haptic mechanotactile feedback sensation, and allowing wear-ability. In particular, the innovation of our Bioinspired Adaptable Multiplanar Haptic (BAMH) system lies in a device that:

1. is able to provide both mechanotactile/steady as well as vibrotactile stimuli with variable intensity over a wide range of frequencies. Hence, the device can stimulate mechanoreceptors including the SAI (sensitive to steady stimulus), the SII (sensitive to vibrotactile stimulus with a frequency lower than 5 Hz), the FAI (sensitive to vibrotactile stimulus with a frequency between 5 Hz and 50 Hz), and the FAII (sensitive to vibrotactile stimulus with a frequency between 40 Hz and 400 Hz).
2. is capable of providing simultaneous stimuli on several planar areas of the entire fingertip surface, i.e., the frontal, lateral, and bottom areas of the finger.
3. allows a level of personalised design, adapting to the fingers' curvature. Haptic feedback is given through pressurised air actuation from a control subsystem to the tethered soft fingertip interface.

Sec. **Future work**, p. 7: "Future work surrounding our haptic device includes broadening the frequency range of vibrotactile feedback, aiming to reach 400 Hz (thereby expanding the stimulation range for FA II mechanoreceptors in our system), as well as maximising the spatial resolution and intensity. This exploration aims to improve touch simulation fidelity while maintaining the versatility and intrinsic safety of our BAMH system. Enhancements in this area will widen the fields of application, for instance, to those requiring lower intensity interactions, e.g., during manipulation activities (in Virtual Reality environments) as well as drilling and those combining haptic and braille technologies [7]. Additionally, understanding individual mechanoreceptor functions is crucial, as is integrating additional feedback modalities like temperature, social context, and shear forces to enhance user immersion and experience. "

Comment 3

The motivation for some design choices could be provided, for instance: why pneumatic actuation was employed? How was the material for the interface chosen?

Thanks for your comment. The BAMH system is a pneumatically actuated, soft material robotic interface. The combination of a soft material, silicone-based structure with air actuation was chosen because of the following design decisions:

- a) Implementation of a soft-materials approach facilitates redesign of moulds to manufacture personalised devices [8], hence, enhances adaptation to fingertip contours.
- b) For fluidic actuation, an extrinsic method, any pressure regulators and supply of pressurised fluid is separated from the haptic feedback interface itself, hence, allowing wear-ability of the haptic interface [5].

- c) In particular, pneumatic actuation offers a lighter alternative to hydraulic actuation, thereby enhancing the wearable nature of the system [5].
- d) Soft material, pneumatic actuated robots are inherently regarded as safe [9], thus enabling the delivery of painless mechanotactile stimuli to the skin through the inflation of a soft membrane with low pressure (≤ 150 kPa).

Material properties, such as elasticity, influence the tactile stimuli (i.e., the stimulus' maximum force intensity, force resolution, and maximum vibration frequency) [10]. Therefore, the material should be chosen depending on the application. In our case, we chose to measure human stimuli sensitivity and differentiation thresholds. The commercially available material Dragon Skin™ Smooth-On with known characteristics [11] was selected for the fingertip interface because:

- a) the softness of the material permits soft surfaces to conform well to the object's surface that they are interacting with (e.g., grasped object or the curvature of the finger) [12].
- b) it allows our BAMH system to stimulate with low(er) internal pressure in the chamber (< 0.2 MPa) than using Dragon Skin™ 30 Smooth On.
- c) its material properties (e.g., elasticity) allow the BAMH system to provide several stimulus intensities below and above the human sensitivity threshold. With Dragon Skin™ 10 SmoothOn, the minimum pressure the BAMH system can deliver inflates enough of the fingertip's chamber membrane to provide a recognisable stimulus for human participants.

The mould used for manufacturing the fingertip interface is based on a human index finger to enhance the adaptability of the fingertip interface to the human finger shape.

We used a 0.5 mm thickness for the chamber's membrane resulting from empirical experience of our manufacturing methods. It is worth noting that a thicker membrane would require higher internal pressure to inflate and stimulate the skin.

Based on the results we obtained through manual stimuli sensitivity and two-point differentiation experiments, our haptic feedback devices required a chamber diameter of less than 2 mm. Hence, a diameter of 1.4 mm was feasible and in line with similar haptic interfaces [13, 14].

We now updated the Results Section as follows:

Sec. **Results**, p. 2: "The BAMH system is a pneumatically actuated, soft material robotic interface. The combination of a soft material, silicone-based structure with air actuation offers a number of benefits:

- For fluidic actuation, an extrinsic method, any pressure regulators and supply of pressurised fluid is separated from the haptic feedback device itself [10], hence, allowing wearability of the haptic device [5].
- In particular, pneumatic actuation offers a lighter alternative to hydraulic actuation, thereby enhancing the wearable nature of the system [5].
- Soft material, pneumatic actuated robots are inherently regarded as safe [9], thus enabling the delivery of painless mechanotactile stimuli to the skin through the inflation of a soft membrane with low pressure (≤ 150 kPa).
- Implementation of a soft-materials approach facilitates redesign of moulds to manufacture personalised devices [8] and, hence, enhances adaptation to the contours of the fingertip

or other body parts [12]. On the other hand, soft material properties, such as hardness, allow further modification of the characteristics of the provided haptic stimuli depending on the applications [11]. ”

Sec. **Results**, p. 3: ”The selection of Dragon Skin™ 20 Smooth On for the fingertip interface was driven by the material properties’ impact on tactile stimuli and the need to accommodate human sensitivity thresholds. Its softness enhances adaptation to finger curvature, while its lower internal pressure requirement compared to other materials facilitates stimulation. A 0.5 mm membrane thickness was chosen by the empirical experience of our manufacturing methods. It is worth noting that a thicker membrane would require higher internal pressure to inflate and stimulate the skin. Based on the results we obtained through manual stimuli sensitivity and two-point differentiation experiments, our haptic feedback devices required a chamber diameter of less than 2 mm. Hence, a diameter of 1.4 mm was feasible considering similar haptic interfaces [13, 14].”

Comment 4

In the assessment section, a series of data regarding sensitivity, differentiation, and classification are presented and discussed. It is not clear in my opinion if the aim of this part of the paper is to assess the capabilities of the proposed device or to evaluate such information about tactile perception in humans.

Response: We thank the Reviewer for their comment. We like to clarify that the contribution of our work lies in development of a haptic device that

1. is able to provide both mechanotactile/steady as well as vibrotactile stimuli with variable intensity over a wide range of frequencies. Hence, the device can stimulate mechanoreceptors including the SAI (sensitive to steady stimulus), the SII (sensitive to vibrotactile stimulus with a frequency lower than 5 Hz), the FAI (sensitive to vibrotactile stimulus with a frequency between 5 Hz and 50 Hz), and the FAII (sensitive to vibrotactile stimulus with a frequency between 40 Hz and 400 Hz).
2. is capable of provide simultaneous stimuli on several planar areas of the entire fingertip surface, i.e., the frontal, lateral, and bottom areas of the finger.
3. allows a level of personalised design, adapting to the fingers’ curvature. Haptic feedback is given through pressurised air actuation from a control subsystem to the tethered soft fingertip interface.

Through this innovation, our system can play an important role in further understanding human touch. Therefore, we performed human stimuli sensitivity and differentiation experiments and we also assessed the stimuli classification to evaluate the capability of the BAMH system to deliver mechano-vibrotactile, variable intensity, simultaneous, multiplanar and operator agnostic stimuli.

To address the Reviewer’s comment and clarify the contribution of our work, we now thoroughly revised the abstract and introduction as follows:

Sec. **Abstract**, p. 1: ”Here, we introduce the Bioinspired Adaptable Multiplanar Haptic (BAMH) system, offering mechanotactile/steady and vibrotactile stimuli with adjustable intensity (up to 298.1 mN) and frequencies (up to 130 Hz). This system can deliver simultaneous stimuli across multiple fingertip areas, adapting to its curvature. The paper includes a full

characterisation of our system. As the device can play an important role in further understanding human touch, we performed human stimuli sensitivity and differentiation experiments to evaluate the capability of delivering mechano-vibrotactile, variable intensity, simultaneous, multiplanar and operator agnostic stimuli.”

Sec. **Introduction**, p. 1: ”Here, we show that our work contributes to enhancing the fidelity of touch feedback, providing a frequency range of haptic mechanotactile feedback sensation, and allowing wear-ability. In particular, the innovation of our Bioinspired Adaptable Multiplanar Haptic (BAMH) system lies in a device that:

1. is able to provide both mechanotactile/steady as well as vibrotactile stimuli with variable intensity over a wide range of frequencies. Hence, the device can stimulate mechanoreceptors including the SAII (sensitive to steady stimulus), the SAI (sensitive to vibrotactile stimulus with a frequency lower than 5 Hz), the FAI (sensitive to vibrotactile stimulus with a frequency between 5 Hz and 50 Hz), and the FAII (sensitive to vibrotactile stimulus with a frequency between 40 Hz and 400 Hz).
2. is capable of provide simultaneous stimuli on several planar areas of the entire fingertip surface, i.e., the frontal, lateral, and bottom areas of the finger.
3. allows a level of personalised design, adapting to the fingers’ curvature. Haptic feedback is given through pressurised air actuation from a control subsystem to the tethered soft fingertip interface.

The stimulus’ intensity range and vibrotactile frequency range are evaluated through the characterisation of the BAMH system. In addition, we performed human stimuli sensitivity and differentiation experiments to evaluate the capability of the BAMH system to deliver mechano-vibrotactile, variable intensity, simultaneous, multiplanar and operator agnostic stimuli, as we believe that our system can play an important role in further understanding human touch.

Comment 5

According to the introduction of the paper, the aim of the experiments should be the assessment of the device. Still, in this case, the obtained results should be compared with some reference data or other devices.

Response: Thank you for the comment. To evaluate our BAMH system, we conducted two sets of experiments. Firstly, we verified the system in the Results Section, in particular, in the BAMH system characterization subsection. Secondly, we performed experiments involving participants to demonstrate our contribution, focusing on understanding distal phalanx sensitivity and stimuli differentiation as detailed in the Distal Phalanx Sensitivity and Distal Phalanx Stimuli Differentiation subsections within the Results Section. These results have been compared to the gold standard of stimulus sensitivity and two-point differentiation, which are the von Frey hairs and a two-point discriminator, illustrated in Figs 3 and 4, respectively.

We now compared the characterization results of our BAMH system to haptic feedback systems from the literature. In particular, we compared performance indicators, including materials, type of stimuli, frequency range, force range, areas of stimulation and simultaneous stimulation across multiple areas. A comparison Table 1 with a detailed explanation has now been added to the Results section.

Table 1 : BAMH’s system results compared to literature’s haptic feedback systems that provide tactile feedback

Haptic systems	Haptic interface material	Mechano-Vibrotactile stimulus	Freq. Range Hz	Max. Force mN	Areas		Simultaneous stimuli across areas
					Lat	Bot Frnt	
Our BAMH system	Dragon Skin™ 20	M, V	0 - 130	298.1	Y	Y	Y
Pin display [15]	Rigid	M**	-	400	Y	Y	Y
SPA-skin [16]	Sylgard 184	M*, V	0 - 100	1000	N	Y	N
SPA-skin [4]	Dragon Skin™ 30	M*, V	0 - 120	300	N	Y	N
Fuppeteer [17]	Rigid	M	-	2100	Y	Y	N
FingerPrint [10]	Flexible 80A	M*, V	1 - 64	7,000	N	Y	N
HAXEL [18]	Various flexible	M, V	~10 - 320	300	N	Y	N

* Denotes that it is our interpretation based on the information provided in the corresponding paper.

** In this work, only steady/mechano tactile feedback is used. Nevertheless, the system has an update time of 50 Hz, so it may be capable of delivering mechanovibrotactile stimuli with a frequency lower than 50 Hz.

Sec. **Results**, p. 4: "When comparing the results of our BAMH system with existing haptic feedback systems providing tactile feedback, as outlined in Table 1, it can be observed that our system, by using Dragon Skin™ 20 as material for the soft fingertip interface can deliver mechano-vibrotactile, with a maximum vibrotactile frequency of 130 Hz. Our system is the only one capable of stimulating both Slow Adapting mechanoreceptors (sensitive frequency range between 0 Hz to $<\sim 5$ Hz) and Fast Adapting mechanoreceptors (sensitive frequency range between ~ 5 Hz to ≥ 400 Hz) with a frequency exceeding 120 Hz. With this capability, our system may stimulate FAII mechanoreceptors at 128 Hz, wherein FAII exhibits the highest peak response when the stimulus induces ≥ 6 μm skin deformation[19]. However, the maximum force output of our system was recorded at 298.1 mN because we were focused on the application of detecting stimuli sensitivity and differentiation thresholds. So, we have capped the maximum internal pressure of our system at 178.2 kPa instead of the 300 kPa achievable by the pressure regulators. Moreover, our bioinspired fingertip interface facilitates the delivery of stimuli across the lateral, bottom, and frontal areas of the finger with simultaneous actuation across and between these areas."

Comment 6

Some results are presented in a rather synthetic way: for instance, a differentiation among sex is identified and discussed, but I think that this difference could be motivated by different aspects that probably are out of the scope of the paper, and would require deeper investigations and more subjects. I would therefore avoid the discussion on the influence of the subject sex in this context.

Response: We like to thank the Reviewer for their comment. In the manuscript, we now revised the results, discussion and corresponding figures to avoid the discussion on the influence of sex. Furthermore, we have included the sex influence as part of future work.

Sec. **Future work**, p. 7: "We envision studies on how sex, environment/social context, and stimulus' location, type, intensity and application time affect the stimuli perception."

Comment 7

Reference:

The overall review of the state of the art is proper in my opinion. I liked the initial introduction to the sense of touch, which is clear and synthetic. As I highlighted in the previous point, I suggest clarifying which gaps and limits of the state-of-the-art have been solved by the developed device and which ones are still open.

Response: Many thanks again for your kind comments. We now addressed your suggestion in our response to your Comment and revised the introduction and future work sections accordingly.

Reviewer 2

Comment 1

Authors developed a pneumatic multiplanar display that can present stimuli up to 298.1 mN in intensity and 130Hz vibration. The authors evaluated the absolute and discrimination threshold of the force.

The strength of this study is to develop a system that can provide spatially distributed vibration up to 130Hz.

Response: We like to thank the Reviewer for acknowledging our work and contribution.

Comment 2

While the Pacinian-FAII's most sensitive frequency is over 200Hz [3], the developed display cannot present such frequencies. In our daily lives, we recognize objects' haptic properties and perform haptic interactions with objects using vibrotactile information containing over 200Hz [20]. If researchers aim to develop a display for application scenarios like those mentioned by the authors (such as training and VR, etc.) that aim to simulate everyday haptic interactions, the display should naturally need to support this frequency range.

[3] Jones, L. A., & Lederman, S. J. (2006). *Human hand function*. Oxford University Press.

[20] Shao, Y., Hayward, V., & Visell, Y. (2016). Spatial patterns of cutaneous vibration during whole-hand haptic interactions. *Proceedings of the National Academy of Sciences*, 113(15), 4188-4193.

Response: We thank the reviewer for this valuable comment and we would like to comment, firstly, on the ability of our device to stimulate the Pacinian-FAII receptors and, secondly, on potential applications including the simulation of everyday interactions.

The work by Jones et al. [3] states that "FA II units are maximally sensitive to vibrotactile temporal frequencies ranging from 40 Hz and > 500 Hz, with a maximum sensitivity around 300 Hz". This is in line with previous research [21, 19], suggesting a frequency range from 40 Hz and > 400 Hz. The response of FA II for stimulus with different intensities and frequencies (illustrated in Fig. 1) also demonstrates that the higher the stimulus intensity, the more the peak FA II response move towards lower frequencies [19]. Therefore, FA II can be stimulated with stimulus ≤ 128 Hz (see FA II peak response illustrated in Fig. 1 with a blue line) and intensity that produces a peak to peak skin displacement $\geq 6 \mu\text{m}$ (where ~ 20 mN produce $\sim 100 \mu\text{m}$ compressive skin displacement [22]). Two of the characteristics of our BAMH system are the ability to provide both mechanotactile/steady as well as vibrotactile stimuli with a wide range of intensities above the human sensitivity threshold and over a wide range of frequencies. Hence, our haptic device can stimulate all four human mechanoreceptors including the FAII receptors, sensitive to vibrotactile stimulus with a frequency between 40 Hz and 400 Hz, as it can provide vibrotactile feedback with an intensity up to 298.1 mN between 40 Hz and 130 Hz.

In response to the reviewer's comment regarding the vibrotactile information present in everyday interactions across frequencies of 200 Hz, we would like to refer to previous research [21, 19, 22, 23, 24]. During routine activities at home, work, and outdoors — such as personal

Editorial Note: Figure below adapted with permission from Roland S Johansson, Ulf Landstro, Ronnie Lundstro, et al. Responses of mechanoreceptive afferent units in the glabrous skin of the human hand to sinusoidal skin displacements. *Brain research*, 244(1):17–25, 1982. Copyright © 1982 Published by Elsevier B.V. [https://doi.org/10.1016/0006-8993\(82\)90899-X](https://doi.org/10.1016/0006-8993(82)90899-X)

Figure 1 **RAII’s response for different intensities and frequency stimulus (taken from [19])**. Each line corresponds to the number of impulses per cycle across stimuli with the same intensity but different frequencies. The intensity is expressed in dB as an amplitude ratio to a 1 mm peak to peak skin displacement stimulus. Therefore, -60 dB corresponds to $1\ \mu\text{m}$ peak to peak skin displacement stimulus. The blue line highlights that FA II high sensitivity response can be achieved with high intensity stimulus (producing a $\geq 6\ \mu\text{m}$ skin deformation).

grooming, slicing a cucumber, plugging in an electrical socket, writing, unlocking a door with a key, and buttoning a shirt — our fingers experience average maximum forces exceeding $1\ \text{N}$ [23, 24]. Considering that the frequency of FAII impulses peaks at $128\ \text{Hz}$ (as indicated by the blue line in Fig. 1) with a skin displacement of at least $6\ \mu\text{m}$ [19] and that the force required to compress the finger’s skin approximately $100\ \mu\text{m}$ is about $20\ \text{mN}$ [22], we can infer that our BAMH system is capable of simulating some everyday haptic interactions. For instance, by delivering with our BAMH system high intensity stimuli to stimulate FAII mechanoreceptors within the frequency range of $40\ \text{Hz}$ to $130\ \text{Hz}$.

In considering the potential applications of our BAMH system in areas like virtual reality (VR) and training, prior research [5] suggests that haptic devices for augmented reality (AR) and VR environments should integrate various types of tactile feedback and utilize interfaces that are soft, thin, and lightweight. Progressing toward this objective, our BAMH system delivers mechano-vibrotactile stimuli across the entire finger within the sensitive frequency range of both the slow-adapting (SA) and fast-adapting (FA) mechanoreceptors, through a soft and lightweight interface.

We acknowledge that our BAMH system currently does not stimulate FAII mechanoreceptors within their entire sensitive frequency range, specifically between $131\ \text{Hz}$ and $> 400\ \text{Hz}$. As a result, we have stated in our paper that future research will focus on expanding the frequency range for vibrotactile feedback to frequencies $> 400\ \text{Hz}$. This expansion will enable the use of our haptic feedback device in applications requiring low stimulus intensity (generating skin displacement lower than $6\ \mu\text{m}$) when interacting with objects through handheld devices.

We have now revised the Introduction and Future Work Sections accordingly:

Sec. **Introduction**, p. 1: "FAII are sensitive to micrometric deformations and vibrotactile stimuli in the range of 40 Hz to >400 Hz [21, 19, 3]. Moreover, as stimuli intensity increases, there is a corresponding increase in the FAII peak response at lower frequencies. To clarify, when the stimulus induces a peak-to-peak skin displacement of $\geq 6 \mu\text{m}$ at a frequency of ≤ 128 Hz, it triggers a higher FAII peak response compared to frequencies exceeding ≤ 128 Hz [19].

Sec. **Future work**, p. 7: "Future work surrounding our haptic device includes broadening the frequency range of vibrotactile feedback, aiming to reach 400 Hz (thereby expanding the stimulation range for FA II mechanoreceptors in our system), as well as maximising the spatial resolution and intensity. This exploration aims to improve touch simulation fidelity while maintaining the versatility and intrinsic safety of our BAMH system. Enhancements in this area will widen the fields of application, for instance, to those requiring lower intensity interactions, e.g., during manipulation activities (in Virtual Reality environments) as well as drilling and those combining haptic and braille technologies [7]. Additionally, understanding individual mechanoreceptor functions is crucial, as is integrating additional feedback modalities like temperature, social context, and shear forces to enhance user immersion and experience."

Comment 3

The haptic displays that performed multiplanar actuation have been developed [25, 15]. This is inconsistent with the authors' note "these current haptic tactile solutions are mainly focused on stimulating the pulp of the fingers and have limited or no simultaneous multiplanar actuation." I have another concern about the sparse spatial resolution of the author's display. The spatial resolution is important in this type of finger-mounted display because the spatial threshold of the fingertip is about 2-3mm.

[25] Taniguchi, T., Sakurai, S., Nojima, T., & Hirota, K. (2018). Multi-point pressure sensation display using pneumatic actuators. In *Haptics: Science, Technology, and Applications: 11th International Conference, EuroHaptics 2018, Pisa, Italy, June 13-16, 2018, Proceedings, Part II* 11 (pp. 58-67). Springer International Publishing.

[15] Ujitoko, Y., Taniguchi, T., Sakurai, S., & Hirota, K. (2020). Development of finger-mounted high-density pin-array haptic display. *IEEE Access*, 8, 145107-145114.

Response: We thank the Reviewer for the comments regarding our statement about current haptic tactile solutions and the spatial resolution of our fingertip interface.

In relation to our statement about current haptic tactile solutions primarily targeting finger pulp stimulation with limited or no simultaneous multiplanar actuation, we wish to emphasize that numerous haptic tactile feedback systems focus primarily on stimulating the finger pulp and may offer restricted or no simultaneous stimulation of the frontal, lateral, and bottom areas of the finger. To provide clarity, we have included a table in the Results section comparing the characteristics of our system with established haptic feedback systems. As it can be seen, our system is the only one that presents all the following characteristics: it uses a soft interface, enhancing conformity to the finger's curvature; it can deliver mechano-vibrotactile stimuli within the frequency range of 0 Hz to 130 Hz to stimulate all four mechanoreceptors associated with the sense of touch; and it delivers variable force intensity stimuli, up to 298.1 mN, and simultaneous multi-point stimuli across and within areas. We have therefore made modifications to the Introduction and Results Sections.

Sec. **Introduction**, p. 1: However, a number of gaps persist in the area of haptic device development due to the multifaceted nature of the sense of touch, influenced by various factors such as social context, temperature, shear force exertion, and characteristics of tactile stimuli [1, 2, 3, 4]. Existing gaps include [5, 6] challenges enhancing levels of realism and fidelity in haptic devices, integrating multimodal sensations effectively, encompassing mechanotactile/steady and vibrotactile feedback, or temperature variations, and miniaturising both the overall interface and its actuators to minimise user distraction and optimise focus on the tactile experience [5, 6]. Finding solutions to these challenges is essential for advancing haptic technologies and unlocking their full potential across various applications.

Sec. **Results**, p. 4: "When comparing the results of our BAMH system with existing haptic feedback systems providing tactile feedback, as outlined in Table 1, it can be observed that our system, by using Dragon Skin™ 20 as material for the soft fingertip interface can deliver mechano-vibrotactile, with a maximum vibrotactile frequency of 130 Hz. Our system is the only one capable of stimulating both Slow Adapting mechanoreceptors (sensitive frequency range between 0 Hz to $<\sim 5$ Hz) and Fast Adapting mechanoreceptors (sensitive frequency range between ~ 5 Hz to ≥ 400 Hz) with a frequency exceeding 120 Hz. With this capability, our system may stimulate FAII mechanoreceptors at 128 Hz, wherein FAII exhibits the highest peak response when the stimulus induces ≥ 6 μm skin deformation[19]. However, the maximum force output of our system was recorded at 298.1 mN because we were focused on the application of detecting stimuli sensitivity and differentiation thresholds. So, we have capped the maximum internal pressure of our system at 178.2 kPa instead of the 300 kPa achievable by the pressure regulators. Moreover, our bioinspired fingertip interface facilitates the delivery of stimuli across the lateral, bottom, and frontal areas of the finger with simultaneous actuation across and between these areas."

Sec. **Results**, p. 5: see Table 1 on next page.

Table 1 : BAMH’s system results compared to literature’s haptic feedback systems that provide tactile feedback

Haptic systems	Haptic interface material	Mechano-Vibrotactile stimulus	Freq. Range Hz	Max. Force mN	Areas		Simultaneous stimuli across areas
					Lat	Bot Frnt	
Our BAMH system	Dragon Skin™ 20	M, V	0 - 130	298.1	Y	Y	Y
Pin display [15]	Rigid	M**	-	400	Y	Y	Y
SPA-skin [16]	Sylgard 184	M*, V	0 - 100	1000	N	Y	N
SPA-skin [4]	Dragon Skin™ 30	M*, V	0 - 120	300	N	Y	N
Fuppeteer [17]	Rigid	M	-	2100	Y	Y	N
FingerPrint [10]	Flexible 80A	M*, V	1 - 64	7,000	N	Y	N
HAXEL [18]	Various flexible	M, V	~10 - 320	300	N	Y	N

* Denotes that it is our interpretation based on the information provided in the corresponding paper.

** In this work, only steady/mechano tactile feedback is used. Nevertheless, the system has an update time of 50 Hz, so it may be capable of delivering mechanovibrotactile stimuli with a frequency lower than 50 Hz.

In regard to the limited spatial resolution of our haptic feedback system, we first like to apologise in not being clear about our contribution. We now revised our contribution which lies in a device that:

1. is able to provide both mechanotactile/steady as well as vibrotactile stimuli with variable intensity over a wide range of frequencies. Hence, the device can stimulate mechanoreceptors including the SAI (sensitive to steady stimulus), the SAI (sensitive to vibrotactile stimulus with a frequency lower than 5 Hz), the FAI (sensitive to vibrotactile stimulus with a frequency between 5 Hz and 50 Hz), and the FAII (sensitive to vibrotactile stimulus with a frequency between 40 Hz and 400 Hz).
2. is capable of provide simultaneous stimuli on several planar areas of the entire fingertip surface, i.e., the frontal, lateral, and bottom areas of the finger.
3. allows a level of personalised design, adapting to the fingers' curvature. Haptic feedback is given through pressurised air actuation from a control subsystem to the tethered soft fingertip interface.

Characterization experiments have been performed to provide sufficient evidence that our contributions have been met. For enhancing clarity, we moved the BAMH characterisation results from the Methods Section to the beginning of the Results Section.

As we believe that our system can play an important role in further understanding human touch, we also performed human stimuli sensitivity and differentiation experiments to evaluate the capability of the BAMH system to deliver mechano-vibrotactile, variable intensity, simultaneous, multiplanar and operator agnostic stimuli. These experimental results align with the 2 mm threshold mentioned and used in [25], reference provided by the reviewer, and have been compared to the gold standard of two-point differentiation, which are the two-point discriminator (see Fig. 4). We understand that these results show a minimum distance of 2 mm, i.e., that the spatial resolution of the device used for these experiments might be on the borderline. However, it is important to note that the spatial resolution of our device, following our design method, can be adjusted below 2 mm, depending on the specific application. This adjustment is achieved by modifying the diameter of the corresponding pillar in the mould, which forms the chamber, and also by reducing the chambers' edge-to-edge distance.

To provide clarity, we have made modifications to the Methods and Future Work Sections.

Sec. **Methods**, p. 2: " Based on the results we obtained through manual stimuli sensitivity and two-point differentiation experiments, our haptic feedback device required a chamber diameter of less than 2 mm. Hence, a diameter of 1.4 mm was feasible considering similar haptic interfaces [13, 14]."

Sec. **Future work**, p. 7: Future work surrounding our haptic device includes ..., as well as maximising the spatial resolution and intensity. This exploration aims to improve touch simulation fidelity while maintaining the versatility and intrinsic safety of our BAMH system.

Comment 4

The purpose of the perceptual experiments is not clearly stated. Initially, it seemed like the aim was to compare the thresholds of real stimuli (von Frey hairs) and those produced by the display to demonstrate a similarity in trends. However, if that

were the case, it would be better to examine the correlation between the thresholds of both. Why were sex, area, and threshold variations across different fingers compared, and how did this lead to evaluating haptic display?

Response: Response: We like to thank the Reviewer for their comment and apologies for the confusion.

Firstly, We now clarified and revised the contribution of our work, which lies in a device with the following criteria:

1. The haptic device is able to provide both mechanotactile/steady as well as vibrotactile stimuli with variable intensity over a wide range of frequencies. Hence, the device can stimulate mechanoreceptors including the SAI (sensitive to steady stimulus), the SAI (sensitive to vibrotactile stimulus with a frequency lower than 5 Hz), the FAI (sensitive to vibrotactile stimulus with a frequency between 5 Hz and 50 Hz), and the FAII (sensitive to vibrotactile stimulus with a frequency between 40 Hz and 400 Hz).
2. The device is capable of provide simultaneous stimuli on several planar areas of the entire fingertip surface, i.e., the frontal, lateral, and bottom areas of the finger.
3. A level of personalised design allows adaptation to the fingers' curvature. Haptic feedback is given through pressurised air actuation from a control subsystem to the tethered soft fingertip interface.

To assess our haptic device, we conducted two sets of experiments: On the one hand, we included the characterisation of the system in the Section Results, in particular, see Section Results: BAMH system characterisation. On the other hand, we carried out experiments with participants to show that we have achieved our contribution demonstrating an area of application in understanding the distal phalanx sensitivity and stimuli differentiation in Section Results: Distal phalanx sensitivity and Distal phalanx stimuli differentiation. These results then are compared to the gold standard of stimulus sensitivity and two-point differentiation using von Frey hairs and a two-point discriminator in Figs 3 and 4.

The latter experiments with human participants allowed us then to further investigate human touch. We evaluated the variation across areas because it is known that we have different distributions of the four mechanoreceptors across the fingers' areas. The threshold variation across fingers allowed us to assess to what extent the fingertip interface is adaptable. The assessment criteria here include accuracy (the fraction of samples that the participants correctly classified), precision (the fraction of samples classified as two-point stimuli that were actually two-point stimuli), classification sensitivity (the fraction of all two-point stimuli samples that were correctly classified), and false positive rate (the probability of false alarm). Following Reviewer's 1 advice, we have removed sex as part of the assessment criteria. Correlation was not used because it expresses the extent to which two variables are linearly related.

Through these experiments, we demonstrate that our BAMH system provides operator-agnostic data, addressing limitations of handheld devices that might add variability to the way that the experiments are carried out and, hence, the data (e.g., due to the limited capability of applying stimulus in the same place, touching the skin with all or part of the transversal areas of the von Frey hairs, and limited capability of applying simultaneous and similar intensity stimuli with two-points discriminator). Therefore, due to these sources of variation of handheld devices, its data will not necessarily be linearly related to the data acquired using the BAMH system. The two-point differentiation test highlighted that the stimuli differentiation of the thumb is

different from that of the other fingers (ring, middle and index). Hence, it is important to manufacture a fingertip inspired by the thumb for examining the thumb. Our BAMH system data showed no statistical difference across fingers for stimuli differentiation. This can be explained by the fact that we used the same fingertip interface for all the fingers.

To add these clarifications, we now revised the contribution, introduction and results sections:

Sec. **Introduction**, p. 1: "Here, we show that our work contributes to enhancing the fidelity of touch feedback, providing a frequency range of haptic mechanotactile feedback sensation, and allowing wear-ability. In particular, the innovation of our Bioinspired Adaptable Multiplanar Haptic (BAMH) system lies in a device that:

1. is able to provide both mechanotactile/steady as well as vibrotactile stimuli with variable intensity over a wide range of frequencies. Hence, the device can stimulate mechanoreceptors including the SAI (sensitive to steady stimulus), the SII (sensitive to vibrotactile stimulus with a frequency lower than 5 Hz), the FAI (sensitive to vibrotactile stimulus with a frequency between 5 Hz and 50 Hz), and the FAII (sensitive to vibrotactile stimulus with a frequency between 40 Hz and 400 Hz).
2. is capable of provide simultaneous stimuli on several planar areas of the entire fingertip surface, i.e., the frontal, lateral, and bottom areas of the finger.
3. allows a level of personalised design, adapting to the fingers' curvature. Haptic feedback is given through pressurised air actuation from a control subsystem to the tethered soft fingertip interface."

Sec. **Introduction**, p. 1: "The stimulus' intensity range and vibrotactile frequency range are evaluated through the characterisation of the BAMH system. In addition, we performed human stimuli sensitivity and differentiation experiments to evaluate the capability of the BAMH system to deliver mechano-vibrotactile, variable intensity, simultaneous, multiplanar and operator agnostic stimuli, as we believe that our system can play an important role in further understanding human touch."

Sec. **Results**, p. 5: "The discrepancy regarding the change of stimuli differentiation across fingers between the BAMH results and those from the two-point discriminator can be explained by the difference in height, length and width between the thumb and the rest of the fingers (Kruskal Wallis, $n_{Thumb} = 18$, $n_{Other\ fingers} = 54$, $p_{h_f} = 5.16 \times 10^{-4}$, $p_{l_f} = 4.71 \times 10^{-8}$, and $p_{w_f} = 3.4 \times 10^{-9}$). Consequently, it is necessary to use a modified fingertip interface to enhance the evaluation of the thumb's stimuli differentiation capabilities."

Comment 5

It's essential to provide information about the vibration waveforms presented to the participant's fingers. While the authors mention driving with PWM signals, the actual waveform presented is unlikely to be a perfect square wave. Perception can be influenced by the shape of the waveform applied to the human skin.

Response: Thanks for your comment. We now added Supplementary Fig. 2, illustrating the waveform of the force applied by the membrane to the human skin for the stimuli sensitivity and differentiation experiments and also the corresponding changes in Section Results.

Sec. **Methods**, p. 8: "Using the fingertip interface, a steady stimulus was applied by actuating one chamber with a duty cycle $\delta = 100\%$ for 1.5 seconds(Supplementary Fig. 1a

Figure 2 **Raw force signal delivered by the BAMH system** a) this steady signal is representative as the stimulus intensity changes across the sensitivity test until the intensity threshold is determined, and b) part of the vibro-tactile stimulus (20Hz 5% duty cycle) used for the differentiation test.

illustrates the raw signal).”

Sec. **Methods**, p. 9: ”Therefore, we chose a PWM signal of 20Hz with a duty cycle of 5% for the valve actuation (Supplementary Fig. 1b illustrates the raw signal).”

Comment 6

In the experiments involving human subjects, the authors restricted the PWM signal frequency to 20Hz. Why was this specific frequency chosen? A study limited to 20Hz would primarily activate FAI, but does this mean FAII can be disregarded?

Response: Thanks for your comment. We now added the characterisation results of the Bioinspired Adaptable Multiplanar mechano-vibrotactile Haptic into Fig. X, with additional information provided in Supplementary Table 2. These results show that the maximum frequency of the vibrotactile stimuli is 130 Hz, when using Dragon Skin™ 20 as material for the fingertip interface. This range is included in the wider range for activating the FAII receptors, ranging from 40 Hz and > 400 Hz.

With regards to the experiments involving human participants, we intend to demonstrate one application area of this device, performing one-point and two-point discrimination. Previous research [26] showed that two-point differentiation distance increases with the frequency of the stimuli. The authors of this study found that the minimum distance for 25 Hz stimuli for two-point differentiation was lower than that for 500 Hz stimuli. Additionally, no statistically significant differences in discrimination distances were found between pulses of 50% duty cycle and those of lower duty cycle. Therefore, by reducing the duty cycle, the power consumption can be reduced with no loss in two-point discrimination capacity [26]. Furthermore, it has been found that in nonhairy skin, the sensitivity threshold is determined by Meissner corpuscles (around 18 Hz stimuli [27]). Therefore, we chose a PWM signal of 20 Hz with a duty cycle of 5% for the valve actuation.

To clarify this, we have now added the following paragraph to the methods and future works sections:

Sec. **Methods**, p. 8: "Previous studies [26, 27] showed that using low-frequency vibrations with a low-duty cycle leads to better spatial differentiation in humans with lower power consumption (there is no statistical difference when using 50% duty cycle than when using low-duty cycle [26]). Therefore, we chose a PWM signal of 20 Hz with a duty cycle of 5% for the valve actuation."

Sec. **Future work**, p. 6: "Future studies are required to obtain insights and further understanding in the area of the human sense of touch. For instance, by working with neuroscientists and social scientists, we can use our BAMH system to investigate individually the components related to this sense and the individual and combined contribution of each mechanoreceptor."

Comment 7

Minor: The second paragraph in the Introduction seems overly detailed. It would be more concise to focus on aspects directly relevant to the study.

Response: Many thanks for this comment. We now revised the second paragraph, focussing on aspects directly linked to the research study. In particular, we removed details related to the receptive field diameter, spatial detail as well as corpuscle concentration per cm²). The new paragraph is as follows:

Sec. **Introduction**, p. 1: "According to the work by Jones [28], the human skin contains myelinated $A\beta$ fibres that respond to mechanical stimuli with the intensity of the stimuli being correlated to their discharge frequency [28] (the frequencies are summarised in Fig. 1-a). These fibres end in Merkel, Meissner, Pacinian, or Ruffini corpuscles. The Merkel-SA1 (Slow Adapting type I) corpuscles in the finger are sensitive to steady force, low frequency ($f < 5$ Hz), dynamic skin deformation [21], and local spatial discontinuities. They have a higher sensitivity to surface features and curvatures. The Meissner-FAI (Fast Adapting type I) corpuscles are four times more sensitive to dynamic skin deformation/motion than the SA1 corpuscles. They can detect sudden forces associated with handheld objects and are responsive to pressures and vibrations between 5 to 50 Hz [21]. The Ruffini-SAI (Slow Adapting type II) corpuscles provide information about the direction of motion or force, particularly when the motion involves skin stretching. They are sensitive to skin stretch and steady forces. The Pacinian-FAII (Fast adapting type II) corpuscles are made to capture large low-frequency stresses and strains encountered in daily manual activities. They have a very low spatial resolution, and respond to distant stimuli [21]. FAII are sensitive to micrometric deformations and vibrotactile stimuli in the range of 40 Hz to >400 Hz [21, 19, 3]. Moreover, as stimuli intensity increases, there is a corresponding increase in the FAII peak response at lower frequencies. To clarify, when the stimulus induces a peak-to-peak skin displacement of ≥ 6 μm at a frequency of ≤ 128 Hz, it triggers a higher FAII peak response compared to frequencies exceeding ≤ 128 Hz [19]. "

References

- [1] Linda M Hollinger and Mary Beth T Buschmann. Factors influencing the perception of touch by elderly nursing home residents and their health caregivers. *International Journal of Nursing Studies*, 30(5):445–461, 1993.

- [2] Stanley J Bolanowski Jr, George A Gescheider, Ronald T Verrillo, and Christin M Checkosky. Four channels mediate the mechanical aspects of touch. *The Journal of the Acoustical society of America*, 84(5):1680–1694, 1988.
- [3] Lynette A Jones and Susan J Lederman. *Human hand function*. Oxford university press, 2006.
- [4] Harshal Arun Sonar, Jian-Lin Huang, and Jamie Paik. Soft touch using soft pneumatic actuator–skin as a wearable haptic feedback device. *Advanced Intelligent Systems*, 3(3):2000168, 2021.
- [5] Jessica Yin, Ronan Hinchet, Herbert Shea, and Carmel Majidi. Wearable soft technologies for haptic sensing and feedback. *Advanced Functional Materials*, 31(39):2007428, 2021.
- [6] C. Wee, K. M. Yap, and W. N. Lim. Haptic interfaces for virtual reality: Challenges and research directions. *IEEE Access*, 9:112145–112162, 2021.
- [7] Christophe Ramstein. Combining haptic and braille technologies: design issues and pilot study. In *Proc. of the Second Annual ACM Conf. on Assistive Technologies*, page 37–44, 1996.
- [8] Gianni Stano and Gianluca Percoco. Additive manufacturing aimed to soft robots fabrication: A review. *Extreme Mechanics Letters*, 42:101079, 2021.
- [9] Sara-Adela Abad, Alberto Arezzo, Shervanthi Homer-Vanniasinkam, and Helge A. Wurdemann. Chapter 4 - soft robotic systems for endoscopic interventions. In Luigi Manfredi, editor, *Endorobotics*, pages 61–93. 2022.
- [10] Zhenishbek Zhakypov and Allison M Okamura. Fingerprint: A 3-d printed soft monolithic 4-degree-of-freedom fingertip haptic device with embedded actuation. In *IEEE 5th Int. Conf. Soft Robotics (RoboSoft)*, pages 938–944, 2022.
- [11] Luc Marechal, Pascale Balland, Lukas Lindenroth, Fotis Petrou, Christos Kontovounisios, and Fernando Bello. Toward a common framework and database of materials for soft robotics. *Soft Robotics*, 2020.
- [12] Jin-Huat Low, Marcelo H. Ang, and Chen-Hua Yeow. Customizable soft pneumatic finger actuators for hand orthotic and prosthetic applications. In *2015 IEEE Int. Conf. on Rehabilitation Robotics (ICORR)*, pages 380–385, 2015.
- [13] Martin Culjat, Chih-Hung King, Miguel Franco, James Bisley, Warren Grundfest, and Erik Dutson. Pneumatic balloon actuators for tactile feedback in robotic surgery. *Industrial Robot: An Int. Journal*, 35(5), August 2008.
- [14] Ge Shi, Andrea Palombi, Zara Lim, Anna Astolfi, Andrea Burani, Silvia Campagnini, Federica GC Loizzo, Matteo Lo Preti, Alessandro Marin Vargas, Emanuele Peperoni, et al. Fluidic haptic interface for mechano-tactile feedback. *IEEE transactions on haptics*, 13(1):204–210, 2020.
- [15] Yusuke Ujitoko, Takaaki Taniguchi, Sho Sakurai, and Koichi Hirota. Development of finger-mounted high-density pin-array haptic display. *IEEE Access*, 8:145107–145114, 2020.
- [16] Harshal A. Sonar, Aaron P. Gerratt, Stéphanie P. Lacour, and Jamie Paik. Closed-loop haptic feedback control using a self-sensing soft pneumatic actuator skin. *Soft Robotics*, 7(1):22–29, 2020.

- [17] Eric M Young and Katherine J Kuchenbecker. Implementation of a 6-dof parallel continuum manipulator for delivering fingertip tactile cues. *IEEE transactions on haptics*, 12(3):295–306, 2019.
- [18] Edouard Leroy, Ronan Hinchet, and Herbert Shea. Multimode hydraulically amplified electrostatic actuators for wearable haptics. *Advanced Materials*, 32(36):2002564, 2020.
- [19] Roland S Johansson, Ulf Landstro, Ronnie Lundstro, et al. Responses of mechanoreceptive afferent units in the glabrous skin of the human hand to sinusoidal skin displacements. *Brain research*, 244(1):17–25, 1982.
- [20] Yitian Shao, Vincent Hayward, and Yon Visell. Spatial patterns of cutaneous vibration during whole-hand haptic interactions. *Proceedings of the National Academy of Sciences*, 113(15):4188–4193, 2016.
- [21] R. S. Johansson and J. R. Flanagan. Coding and use of tactile signals from the fingertips in object manipulation tasks. *Nature Reviews Neuroscience*, 10(5):345–359, 2009.
- [22] Brygida M Dzidek, Michael J Adams, James W Andrews, Zhibing Zhang, and Simon A Johnson. Contact mechanics of the human finger pad under compressive loads. *Journal of The Royal Society Interface*, 14(127):20160935, 2017.
- [23] MM Keller, R Barnes, and C Brandt. Evaluation of grip strength and finger forces while performing activities of daily living. *Occupational Health Southern Africa*, 28(5):187–191, 2022.
- [24] Michael Riddle, Joy MacDermid, Sydney Robinson, Mike Szekeres, Louis Ferreira, and Emily Lalone. Evaluation of individual finger forces during activities of daily living in healthy individuals and those with hand arthritis. *Journal of Hand Therapy*, 33(2):188–197, 2020.
- [25] Takaaki Taniguchi, Sho Sakurai, Takuya Nojima, and Koichi Hirota. Multi-point pressure sensation display using pneumatic actuators. In *Haptics: Science, Technology, and Applications: 11th International Conference, EuroHaptics 2018*, volume 2, pages 58–67, 2018.
- [26] CA Perez, CA Holzmann, and HE Jaeschke. Two-point vibrotactile discrimination related to parameters of pulse burst stimulus. *Med. Biol. Eng. Comput.*, 38(1):74–79, 2000.
- [27] P. J. J. Lamoré and C. J. Keemink. Evidence for different types of mechanoreceptors from measurements of the psychophysical threshold for vibrations under different stimulation conditions. *The Journal of the Acoustical Society of America*, 83(6):2339–2351, 06 1988.
- [28] Edward G. Jones. The sensory hand. *Brain*, 129(12):3413–3420, 12 2006.

REVIEWER COMMENTS

Reviewer #1 (Remarks to the Author):

The paper has been carefully modified taking into account the comments and suggestions that I provided in my previous review. I have no further requests on paper contents. As a very minor comment, if possible, I would revise the figures (e.g. modifying text fonts, the thickness of support lines, etc.) to increase their clarity.

Reviewer #2 (Remarks to the Author):

We thank the authors for revising the manuscript. I have read the revisions and reviewer comments, which have deepened my understanding of this paper. Based on the revisions, I now grasp the core contributions of the paper, which are threefold:

- A. This device covers frequencies up to 130 Hz.
- B. This device stimulates multiple planar surfaces of fingers.
- C. Personalization

However, no experiments involving humans exist for any of A, B, and C, although they are of great importance for elucidating the aforementioned contributions regarding human interface devices. Therefore, I think that the current form of the paper is unable to fully demonstrate that contribution and it is difficult to judge the validity of contributions.

In my opinion, in research on haptic feedback devices, human experiments are essential for objectively assessing the device's performance. Comparing specifications between devices could serve but is not enough. This is because haptic feedback devices are meant to be presented to humans, and evaluations from the perspective of human sensation and perception often reveal new contributions or issues with the device.

To demonstrate the contribution of this paper:

1. Experiments supporting the claim of being able to provide vibrations up to 130 Hz that are effective for humans are necessary. For example, investigating whether the threshold values or variations in threshold values when presenting vibrations around 130 Hz are similar to those found in previous studies. The experimental results of the author's experiments can be reproduced even with a device that can output up to 30Hz.
2. Experiments validating the appropriate stimulation of multiple locations are required.
3. Experiments validating the effectiveness of the authors' proposed personalization methods is required.

Regarding the second point, the authors' comment is: "On the other hand, we carried out experiments with participants to show that we have achieved our contribution demonstrating an area of application in understanding the distal phalanx sensitivity and stimuli differentiation in Section Results: Distal phalanx sensitivity and Distal phalanx stimuli differentiation. These results then are compared to the gold standard of stimulus sensitivity and two-point differentiation using von Frey hairs and a two-point discriminator in Figs 3 and 4."

However, in my view, to examine the functionality of human touch, the validity of the device used in such investigations should be confirmed from the perspective of human perception beforehand. Unfortunately, this is lacking in this paper.

First and foremost, the validity of the device needs to be established. It could be clarified to what extent the device aligns with stimulus sensitivity and two-point discrimination using von Frey hairs, which would substantiate its validity from the perspective of human perception. In this case, quantitatively demonstrating the similarity is necessary. For instance, I propose that authors calculate the correlation between the threshold values obtained using the developed device under

various conditions and those obtained using von Frey hairs.

Regarding authors statement, "Correlation was not used because it expresses the extent to which two variables are linearly related." Does this imply a lack of alignment with the results obtained using von Frey hairs?

Minor:

- Please add the information about raw waveforms from 5Hz to 130Hz with 5Hz interval. In addition, I'd like to see the measured vibration in the frequency domain.

A bioinspired adaptable multiplanar mechano-vibrotactile haptic system (NCOMMS-23-44661-T): Reviewers comments and answers

DBPR

The authors express their gratitude to the Reviewers for dedicating their time to reviewing our work and providing us with insightful comments. We have addressed point-by-point all comments and concerns raised, which we believe has substantially improved the manuscript. Additionally, a version of the paper highlighting the changes made throughout is appended to this response.

Reviewer 1

The paper has been carefully modified taking into account the comments and suggestions that I provided in my previous review. I have no further requests on paper contents. As a very minor comment, if possible, I would revise the figures (e.g. modifying text fonts, the thickness of support lines, etc.) to increase their clarity.

Response: Many thanks to the reviewer for dedicating time to reviewing this article and for offering constructive feedback. Figs. 2, 3, and 6 have been revised (e.g., increasing font size and increasing the thickness of some of the lines) in response to your valuable comments.

Reviewer 2

General comment

We thank the authors for revising the manuscript. I have read the revisions and reviewer comments, which have deepened my understanding of this paper. Based on the revisions, I now grasp the core contributions of the paper, which are threefold:

- A. This device covers frequencies up to 130 Hz.
- B. This device stimulates multiple planar surfaces of fingers.
- C. Personalization.

However, no experiments involving humans exist for any of A, B, and C, although they are of great importance for elucidating the aforementioned contributions regarding human interface devices. Therefore, I think that the current form of the paper is unable to fully demonstrate that contribution and it is difficult to judge the validity of contributions.

In my opinion, in research on haptic feedback devices, human experiments are essential for objectively assessing the device's ability. Comparing specifications between devices could serve but is not enough. This is because haptic feedback devices are meant to be presented to humans, and evaluations from the perspective of human sensation and perception often reveal new contributions or issues with the device.

To demonstrate the contribution of this paper:

1. Experiments supporting the claim of being able to provide vibrations up to 130 Hz that are effective for humans are necessary. For example, investigating whether the threshold values or variations in threshold values when presenting vibrations around 130 Hz are similar to those found in previous studies. The experimental results of the author's experiments can be reproduced even with a device that can output up to 30 Hz.
2. Experiments validating the appropriate stimulation of multiple locations are required.
3. Experiments validating the effectiveness of the authors' proposed personalization methods is required.

Response: We like to thank the Reviewer for acknowledging our work and contributions and for the constructive feedback.

In addition to the experiments performing human stimuli sensitivity and differentiation experiments, we understand that the Reviewer requests additional human participants' experiment to evaluate the capability of our device from the human perspective. In particular, 3 experiments are requested:

1. An experiment to demonstrate that the stimuli frequencies are perceived by a human fingertip and their impact on the perception threshold. This experiment aims supporting our contribution A.

2. An experiment to demonstrate that human participant can appreciate stimulation in multiple planar surfaces. This relates to contribution B.
3. An experiment to illustrate the personalisation possibility of our device and method. To support contribution C.

We have now conducted additional experiments involving human participants as requested, supporting our contributions A and B. An experiment to validate contribution C would meant in our opinion to reproduce all verification and validation experiments presented in the current version of the paper with cohorts of different fingertip sizes. This work would be limited by the time constraints, hence, we moved contribution C to future work.

The remainder of the document is structured so that each experiment and related comments are addressed in separate sections to provide required details.

Experiment 1: Supporting the contribution of being able to provide vibrations up to 130 Hz

Response: As requested, we have now performed experiments to *(i)* support our contribution of our device to be able to provide vibrations up to 130 Hz and *(ii)* that are effective for humans.

(i) The BAMH system’s capability to deliver vibrotactile stimulus up to 130 Hz can be demonstrated through analysing raw data illustrating the force exerted by the BAMH system. For obtaining this data, a vibrotactile stimuli with $\delta = 75\%$ ranging from 5 Hz up to 130 Hz in steps of 5 Hz was used. Supplementary Figs. 3 - 5 presents this data in the time domain and frequency domain.

(ii) We have now evaluated the index distal phalanx sensitivity threshold for stimulus with frequencies up to 130 Hz to demonstrate the need to provide mechano-vibrotactile stimuli up to this range for human fingertips. The experiment was carried out on 14 participants using von Frey hairs and the BAMH system with 0 Hz, also referred to as steady/mechanotactile stimulus, and vibrotactile stimulus with duty cycle $\delta = 75\%$ at frequencies of 2 Hz, 20 Hz, 60 Hz, 90 Hz, and 130 Hz. These frequencies were chosen considering the minimum overlap between the sensitivity frequency range of the four mechanoreceptors mainly associated with the touch (SAI, SAII, FAI, and FAII). The methodology (e.g., timings and pressurisation steps) was repeated in line with the sensitivity test across the fingertips. The experiment was repeated for each area of the index finger in a random order. One sample per area per frequency was taken for each participant.

The results shown in Fig. 4 concerning the evaluation of the index distal phalanx sensitivity across frequencies reveal the following:

1. Human stimulus sensitivity thresholds vary across regions and frequencies, highlighting the necessity of a system capable of delivering multi-planar stimuli with variable intensity and frequency.
2. The sensitivity threshold of the index stimulus is skewed to the right indicating that the highest stimulus intensity threshold occurs at 2 Hz, with the threshold decreasing as the frequency increases (see Fig. 4a).
3. Sensitivity data from von Frey hairs aligns with sensitivity results from the BAMH system, particularly for stimulus frequencies exceeding 60 Hz, as illustrated in Fig. 4b. This correlation highlights the importance of employing frequencies within the sensitive range

Figure 4 — **Index distal phalanx sensitivity.** **a** Sensitivity across areas and BAMH’s stimuli frequencies. The median is highlighted with a notch, + are outliers, and each box’s limits are the second and third quartiles. **b** Linear correlation between the result obtained using the BAMH system and von Frey hairs (handheld device) across different areas of the fingertip. Light blue shading emphasizes coefficients exceeding 0.5, with increasing color intensity indicating proximity to a correlation coefficient of 1 (where 1 represents direct relationship of the variables).

of all four mechanoreceptors associated with touch to gain deeper insights into their individual contributions to touch perception.

These findings confirm previous research [1, 2] indicating that FAs (Meissner and Pacinian corpuscles) are the most sensitive mechanoreceptors in von Frey threshold determination [1]. Hence, a correlation is evident between the results obtained from von Frey hair testing and those from the BAMH system for frequencies exceeding 60 Hz (see Fig. 4b). Additionally, the observed response of Pacinian corpuscles to various frequency stimuli, which is skewed to the right showing that Pacinian corpuscles, are sensitive to high intensity low frequency stimulus and lower intensity higher frequencies stimulus [2], resembles the threshold outcomes obtained through our BAMH system for each area (see Fig. 4a).

The paper has now been updated and strengthened as follows:

Sec. **Results**, p. 6: Fig. 4 **Index distal phalanx sensitivity** was included.

Sec. **Results**, p. 4: Supplementary Figs. 3-5 illustrate the raw force data of mechano-vibrotactile stimulus delivered by the BAMH system.

Sec. **Results**, p. 5: The index finger sensitivity analysis results across frequencies and areas (see Fig. 4a) demonstrate that the sensitivity threshold vary across regions and frequencies. The results across frequencies are skewed to the right, indicating that the highest stimulus intensity threshold occurs at 2 Hz, with the threshold decreasing as the frequency increases. Also, Fig. 4b confirms that the sensitivity data from von Frey hairs correlates with the BAMH system results, in particular, for stimulus frequencies exceeding 60 hertz.

[...] The results also demonstrate that the haptic system can provide stimuli [...] 2) with frequency within the sensitivity range of touch mechanoreceptors [...]. Another finding concerns the sensitivity results across frequencies resembling the Pacinian (FAII) response to various frequency stimulus [2]. This may be explained by previous research [1], demonstrating that FA

mechanoreceptors are the most sensitive receptors in von Frey threshold determination.

Sec. **Methods**, p. 10: *Index finger sensitivity analysis test across frequencies*: This test determines the chamber’s minimal force/internal pressure required for the participant to feel a vibrotactile stimulus with $\delta = 75\%$. The evaluated frequencies were 2 Hz, 20 Hz, 60 Hz, 90 Hz, and 130 Hz. These frequencies were chosen considering the minimum overlap between the sensitivity frequency range of the SA and FA mechanoreceptors. The rest of the methodology was repeated in line with the sensitivity analysis across fingers, previously described.

Experiment 2: Experiments validating the appropriate stimulation of multiple locations supporting contribution B

For this experiment, the reviewer added: **Regarding the second point, the authors’ comment is: ”On the other hand, we carried out experiments with participants to show that we have achieved our contribution demonstrating an area of application in understanding the distal phalanx sensitivity and stimuli differentiation in Section Results: Distal phalanx sensitivity and Distal phalanx stimuli differentiation. These results then are compared to the gold standard of stimulus sensitivity and two-point differentiation using von Frey hairs and a two-point discriminator in Figs 3 and 4.”**

However, in my view, to examine the functionality of human touch, the validity of the device used in such investigations should be confirmed from the perspective of human perception beforehand. Unfortunately, this is lacking in this paper.

First and foremost, the validity of the device needs to be established. It could be clarified to what extent the device aligns with stimulus sensitivity and two-point discrimination using von Frey hairs, which would substantiate its validity from the perspective of human perception. In this case, quantitatively demonstrating the similarity is necessary. For instance, I propose that authors calculate the correlation between the threshold values obtained using the developed device under various conditions and those obtained using von Frey hairs.

Regarding authors statement, ”Correlation was not used because it expresses the extent to which two variables are linearly related.” Does this imply a lack of alignment with the results obtained using von Frey hairs?

Response: As suggested by the Reviewer, experiments with human participants were performed (*i*) to validate the need of appropriate stimulation across areas and (*ii*) to evaluate the correlation between the results obtained using von Frey hairs and our BAMH system.

Experiment 2-i: Validating the need to provide appropriate stimulation of multiple locations using human participants

Participants for Experiment 2 were the same cohort as the participants for Experiment 1. The conducted experiment evaluates the participants’ ability to perceive stimulation at multiple locations. This was done by measuring the participants’ abilities to identify pairs of stimuli, one at the rear of the index finger in areas A, F, and E and one at the front of the index finger in areas B, C, and D. These combinations were chosen to be similar to those used by Ivanov et al. [3] in their assessment of multi-contact actuation. The participants were presented with a sheet showing top views of stimuli pair locations numbered from 1 to 9. Each stimulus was then displayed twice to the participant for training. In line with the stimuli differentiation

analysis/two-point differentiation test, a single point in both areas was actuated simultaneously at a pressure of 154.8 kPa, $f = 20$ Hz, $\delta = 5\%$ for 1.5 s. After training, the stimuli pairs were presented to the participant in a random order, with each stimuli pair being presented twice to give a total of 18 pairs. After each pair of stimuli were presented, the participant was required to identify the stimuli pair number using the sheet.

Figure 6 — **Index stimuli differentiation across areas.** **a** Confusion matrix illustrating the ability of participants to recognize simultaneous stimuli applied in pairs in the frontal and rear area of the fingertip, where the pairs are denoted using the name of the areas, e.g., BA denotes that a simultaneous stimulus was applied in areas B and A. From these pair stimulus predictions, the confusion matrix for correctly identifying the stimulus in **b** the frontal areas and **c** the back areas was obtained. **d** Participants’ prediction accuracy.

The index stimuli differentiation across various areas, as illustrated in Fig. 6, reveals variation in human classification accuracy despite the use of identical vibrotactile stimuli in both regions. Notably, combinations characterized by being aligned along the finger (DE, BA, and CF) exhibited the highest prediction accuracy across trials, with coefficients of 89.3%, 85.7%, and 82.1%, respectively (see Fig. 6a).

Moreover, when comparing participant predictions between frontal and posterior areas (see Fig. 6b/c), it is noteworthy that accuracy in identifying stimuli in posterior areas (ranging from 84.5% in area A to 94% in area F) surpasses that in frontal areas (ranging from 66.7% in area C to 83.3% in area D). This discrepancy may be attributed to the variability observed in index finger sensitivity results (see Fig. 4a), where sensitivity to 20 Hz stimuli exhibits low variability in area F, while area C demonstrates high variability.

This is further confirmed by the accuracy findings presented in Fig. 6d. In particular, the accuracy in identifying rear areas is 88.89%, whereas for frontal areas, it is 75.79%. However, the accuracy decreases to 70.63% when identifying both frontal and rear areas.

These results highlight the effects of stimulus intensity and frequency in the human sensitivity threshold and stimuli differentiation across areas. Consequently, there is a clear requirement for a system capable of delivering independent stimuli across areas.

The paper was updated and strengthened as follows:

Sec. Results, p. 7: Index stimuli differentiation across areas.

The results presented in Fig. 6 reveal variation in human stimuli differentiation accuracy despite using identical vibrotactile stimuli in both regions. Notably, combinations characterized by areas’ alignment along the finger (DE, BA, and CF) exhibited the highest percentage of

correct pair identification across trials, with coefficients of 89.3%, 85.7%, and 82.1%, respectively (see Fig. 6a). Furthermore, participants mainly confused the CA pair with the BA pair and the DE pair with the CE pair. This can be explained by the fact that participants commonly misidentify stimulation in areas B and D as stimulation in area C (see Fig. 6b). It is also worth mentioning that stimuli differentiation in rear areas (ranging from 84.5% in area A to 94% in area F, see Fig. 6c) surpasses that in frontal areas (ranging from 66.7% in area C to 83.3% in area D). This discrepancy may be attributed to the variability observed in index finger sensitivity results (see Fig. 4a), where sensitivity to 20 Hz stimuli exhibits low variability in area F, while area C demonstrates high variability. This is further corroborated by the accuracy findings presented in Fig. 6d. In particular, the participants' prediction accuracy for correctly identifying the pairs is 70.63%. From these results, it was found that participants' accuracy in identifying the rear area of the pair is 88.89%, whereas that for frontal areas is 75.79%.

Sec. **Results**, p. 8: Fig. 6 **Index stimuli differentiation across areas** was included.

Sec. **Methods**, p. 10: *Index finger stimuli differentiation analysis across areas*: This test evaluates the participants' abilities to identify pairs of stimuli, one at the rear of the index finger (areas A, F, and E) and one at the front (areas B, C, and D). These combinations are similar to those used in [3] in their assessment of multi-contact actuation. The stimuli comprised a single point in both areas, actuated simultaneously, at a pressure of 154.8 kPa, $f = 20$ Hz, $\delta = 5\%$ for 1.5 s. The participants were presented with a sheet showing top views of stimuli pair locations numbered from 1 to 9 (see Supplementary Fig. 11). For training, each stimulus was displayed twice to each participant. Then, the stimuli pairs were presented in a random order, with each stimuli pair being presented twice to give a total of 18 pairs. After each pair of stimuli were presented, the participant was required to identify the stimuli pair number using the sheet.

Sec. **Supplementary information**, p. 20: Supplementary Fig. 11 **Index finger stimuli differentiation stimuli pairs** was included.

Experiment 2-ii: Correlation across von Frey hairs and BAMH system results

In this experiment, we evaluated the correlation between the von Frey hairs results and those obtained using the BAMH system. Following the methodology (e.g., timings and pressurisation steps) used to evaluate human sensitivity test across fingertips, we measured the sensitivity threshold of the participants using von Frey hairs for the 14 participants.

As demonstrated in Fig. 4b, the sensitivity outcomes obtained from the index finger using von Frey hair are correlated to those obtained using the BAMH system for vibrotactile stimuli exceeding 60 Hz. This observation is consistent with prior studies (Johansson et al. [1]), which have shown that fast-adapting mechanoreceptors, namely Meissner and Pacinian corpuscles, exhibit high sensitivity during sensitivity threshold determination using von Frey hairs. Regarding the Reviewer's question: "Does this imply a lack of alignment with the results obtained using von Frey hairs?", it is worth looking into results and previous research [1], that demonstrate that the results alignment depends on the frequency of the applied stimulus. So, the results obtained using von Frey hairs correlate to the BAMH's results when vibrotactile stimulus higher than 60 Hz was employed.

The paper was updated and strengthened as follows:

Sec. **Results**, p. 6: Fig. 4b correlation results were included.

Supplementary Figure 11 : Index finger stimuli differentiation stimuli pairs illustration.

Participants were handed this illustration that presents, from the top view, the evaluated nine-pair combinations. The top view was employed to facilitate the association of the received stimulus and those from the illustration. The vibrotactile stimuli characteristics are the same used for the distal phalanx stimuli differentiation (internal pressure of 154.8 kPa, $f = 20$ Hz, $\delta = 5\%$, and simultaneous stimuli application during 1.5 s). The paired areas are: (1) DE, (2) DF, (3) DA, (4) CE, (5) CF, (6) CA, (7) BE, (8) BF, and (9) BA.

Sec. **Results**, p. 5: Fig. 4b confirms that the sensitivity data from von Frey hairs correlates with the BAMH system results, in particular, for stimulus frequencies exceeding 60 Hz

Sec. **Results**, p. 6: This may be explained by results obtained by previous research [1], demonstrating that FAs mechanoreceptors are the most sensitive receptors in von Frey threshold determination.

Minor comment

- Please add the information about raw waveforms from 5Hz to 130Hz with 5Hz interval. In addition, I'd like to see the measured vibration in the frequency domain.

Response: We have incorporated the raw waveforms (in the time and frequency domains) of the force provided by the BAMH system for vibrotactile stimuli ranging from 5 Hz to 130 Hz at intervals of 5 Hz.

The paper was updated as follows:

Sec. **Results**, p. 4: Supplementary Figs. 3 - 5 illustrate the raw force data of the vibrotactile stimulus delivered by the BAMH system.

References

[1] R. S. Johansson, A. B. Vallbo, and G. Westling. Thresholds of mechanosensitive afferents in the human hand as measured with von frey hairs. *Brain Research*, 184(2):343–351, 1980.

- [2] Roland S Johansson, Ulf Landstro, Ronnie Lundstro, et al. Responses of mechanoreceptive afferent units in the glabrous skin of the human hand to sinusoidal skin displacements. *Brain research*, 244(1):17–25, 1982.
- [3] Aysien Ivanov, Daria Trinitatova, and Dzmitry Tsetserukou. Linkring: a wearable haptic display for delivering multi-contact and multi-modal stimuli at the finger pads. In *Haptics: Sci., Technology, Applicat.: 12th Int. Conf., EuroHaptics 2020*, volume 12, pages 434–441, 2020.

Supplementary Figure 3 : Raw force data of the BAMH system in the time and frequency domain for vibrotactile stimulus, with $\delta = 75\%$, between 0 Hz and 50 Hz. The frequency domain plots show the fundamental and the harmonic frequency components, whose frequency is a multiple of that of the force signal in the time domain.

Supplementary Figure 4 : **Raw force data of the BAMH system in the time and frequency domain for vibrotactile stimulus, with $\delta = 75\%$, between 55 Hz and 100 Hz.** The frequency domain plots show the fundamental and the harmonic frequency components, whose frequency is a multiple of that of the force signal in the time domain.

Reviewers' comments:

Reviewer #2 (Remarks to the Author):

First and foremost, I want to express my appreciation for the authors' efforts to revise the paper.

However, upon reviewing the results of the spectral analysis conducted with square waves at various frequencies and a duty ratio of 75%, I've noted some areas of concern.

Primarily, there seems to be a challenge in producing pure variational components. Across all frequencies, the steady components appear to outweigh the variational ones. This raises questions about the device's ability to precisely control the vibration components crucial for human tactile perception. It also casts doubt on the validity of the perception experiments discussed in the paper, particularly those involving threshold assessments for each frequency.

Furthermore, there are noticeable distortions, including the presence of harmonic components and deviations from ideal square waveforms. For instance, in instances like 120Hz, it appears that the steady or 240Hz components are stronger than the 120Hz ones.

These observations might be attributed to various factors within the entire system, including air resistance, as indicated by prior research. Despite the fast valve switching, it's plausible that the overall system isn't operating at such high speeds. As a result, I find myself questioning the reliability of the experimental outcomes, especially concerning frequency-related findings.

In essence, while there's a claim about the system's strength in presenting both steady and variational components up to 130Hz, given the mentioned concerns, making a definitive assertion on this front could be challenging.

Similarly, the assertion regarding multi-point presentation raises some uncertainties, primarily due to the lower point density compared to previous research on multi-point displays. Consequently, asserting the strength of a multi-point (multi-planar) presentation might require further consideration.

A bioinspired adaptable multiplanar mechano-vibrotactile haptic system (NCOMMS-23-44661-T): Comments and answers

DBPR

Reviewer 2

General comment

First and foremost, we want to express our appreciation for the authors' efforts to revise the paper.

Many thanks for acknowledging our effort in revising the paper and addressing your constructive comments thoroughly. We express our gratitude to Reviewer 2 for dedicating the time to reviewing our responses. Based on their latest comments, we have now responded to their remarks in a point-by-point manner. Some of their comments have resulted in substantially improving the manuscript. A version of the paper highlighting the changes made throughout is appended to this response.

Comment 1

However, upon reviewing the results of the spectral analysis conducted with square waves at various frequencies and a duty ratio of 75%, I've noted some areas of concern.

Primarily, there seems to be a challenge in producing pure variational components. Across all frequencies, the steady components appear to outweigh the variational ones. This raises questions about the device's ability to precisely control the vibration components crucial for human tactile perception. It also casts doubt on the validity of the perception experiments discussed in the paper, particularly those involving threshold assessments for each frequency.

Furthermore, there are noticeable distortions, including the presence of harmonic components and deviations from ideal square waveforms. For instance, in instances like 120Hz, it appears that the steady or 240Hz components are stronger than the 120Hz ones.

These observations might be attributed to various factors within the entire system, including air resistance, as indicated by prior research. Despite the fast valve switching, it's plausible that the overall system isn't operating at such high speeds. As a result, I find myself questioning the reliability of the experimental outcomes, especially concerning frequency-related findings.

In essence, while there’s a claim about the system’s strength in presenting both steady and variational components up to 130Hz, given the mentioned concerns, making a definitive assertion on this front could be challenging.

Response: We would like to thank Reviewer 2 for their comment. We apologise for any confusion and like to use this opportunity to provide in-detail clarification to their remarks with regards to: (i) the capability of our system to delivering pure ”variational components”, (ii) the harmonic components of an ideal square wave, and (iii) the validity of the sensitivity threshold assessments across frequencies.

Our response to each of these concerns are as follows:

- (i) **Capability of our system to delivering pure ”variational components”:** Our BAMH system pneumatically inflates the membrane using positive pressure, producing a pulse force signal with a positive offset. Consequently, in the frequency analysis, a 0 Hz component is expected for any type of stimulus. For a signal with a duty cycle of $\delta = 75\%$, the 0 Hz component is, by definition, 0.75 times the pulse amplitude and is known as the DC component or signal average over one period [1].

Our BAMH system, like other haptic feedback systems from the literature [2, 3], including the Pin array displays [4, 5] suggested by Reviewer 2 in the first review comments, do not apply negative forces to the skin. Therefore, these systems also provide stimuli with a DC component higher than zero. For clarity and in line with the frequency analysis methods in [6], kindly provided by Reviewer 2 in the first rounds of reviews, we have removed the DC component via mean subtraction.

- (ii) **Harmonic components of an ideal square wave/pulse signal:** We use the Fast Fourier transform to demonstrate that our BAMH system can deliver pulse vibration stimuli at different frequencies. This is shown by the spectral coefficient at the multiples of the stimuli frequency, f_0 (as shown in Fig. 1b/c/e). The envelope of the Fourier transform, the function $\text{sinc} = \frac{\sin x}{x}$ is shown in Fig. 1a. Subfigures 1b-c illustrate the envelope and spectral coefficients of a pulse signal with frequency f_0 and duty cycle, $\delta = 12.5\%$ and $\delta = 50\%$, respectively. The shorter the duty cycle, the more spectral coefficients in each lobe [1]. However, it is important to note that the Fast Fourier Transform (FFT) accuracy is affected by (a) the time series length, which should be 2^n where $n \in \mathbb{R}$, and (b) the time series containing a whole number of periods of the pulse signal.

When these requirements are not met, the accuracy of the FFT is lower even in the case of a pure pulse signal ($f_0 = 120 \text{ Hz}$, $\delta = 75\%$). This is evidenced by the higher magnitude of the peak of the second spectral coefficient compared to the first one (see Subfigures 1d-e, and the code from the `fft()` function taken from the Matlab 2022a documentation, which is provided at page 5 of this document). For real noisy data, meeting these requirements becomes even more challenging. Hence, in this article, we maintain our discussion and claim about providing pulse stimuli at several frequencies because the spectral coefficients are at the multiples of f_0 . It has to be also mentioned that the FFT returns complex numbers, and we do not consider the phase in our frequency analysis. Therefore, for the frequency analysis, we only illustrate the absolute value of the single-sided amplitude of the spectrum. As a consequence, the negative coefficients in Subfigures 1b-d are illustrated as positive in Subfigure 1e and in our frequency analysis plots in the paper(see Supplementary Figs. 3 - 5).

- (iii) **Validity of the sensitivity threshold assessments across frequencies:**. The validity of our human perception analysis (Fig. 4) for different frequencies, previously requested by Reviewer 2, is supported by the statistical analysis of our results and the fact that our results are in line with previous research [7, 8].

Statistical analysis demonstrates that the sensitivity threshold varies across regions (Kruskal Wallis, $n = 84$, $p = 2.55 \times 10^{-15}$) and frequencies (Kruskal Wallis, $n = 98$, $p = 5.03 \times 10^{-46}$). This indicates that human participants perceive stimulus variations due to frequency changes. Furthermore, the results across frequencies are skewed to the right, indicating that the highest stimulus intensity threshold occurs at 2 Hz (two-sided Wilcoxon rank sum test, 1% significance level, $p \leq 1.58 \times 10^{-11}$), with the threshold decreasing as the frequency increases.

Our human stimulus perception analysis match the findings from literature [7, 8]. The results demonstrate that FAs (Meissner and Pacinian corpuscles) are the most sensitive mechanoreceptors in von Frey threshold determination [7]. Hence, this explains the correlation between the results obtained from von Frey hair testing and those from our BAMH system for frequencies exceeding 60 Hz (see Fig. 4b). Additionally, the observed response of Pacinian corpuscles to various frequency stimuli, presented in [8], resembles the threshold outcomes obtained through our BAMH system for each area (see Fig. 4a).

After conducting a frequency domain and statistical analysis, and observing alignment with previous studies, we can confidently conclude that our BAMH system is capable of delivering vibration stimuli detectable by human participants across the sensitive frequency range of all four mechanoreceptors associated with the sense of touch (SAI, SAII, FAI, and part of the sensitive frequency range of FAII).

The paper was updated and strengthened as follows:

We have now updated the term "vibrotactile stimulus" to "vibrotactile pulse stimulus" in the entire paper to highlight that our system delivers pulse stimulus.

Supplementary Figs. 3 - 5: Please see the updated version at the end of this document. The frequency analysis has been revised by removing the DC component to focus on the frequency components higher than 0 Hz. Additionally, the caption has been updated to: "The frequency domain plots demonstrate the BAMH system's capability to deliver pulse stimulus by exhibiting the spectral coefficients at the frequencies that are a multiple of the stimulus frequency."

Sec. **Results**, p. 4: The spectral coefficients at the frequencies that are a multiple of the pulse stimulus are evidence of the BAMH's system capability to deliver pulse stimulus. Due to inherent limitations in the frequency analysis of real noisy data (e.g., the evaluated time series not containing complete periods of the pulse, see Methods), the magnitude of the spectral components is not considered in the analysis.

Sec. **Methods**, p. 11: The frequency analysis of the force utilises the Fast Fourier Transform (FFT) of the force-time series data. The FFT was obtained using the FFT function in Matlab, where the DC component was removed by mean subtraction, consistent with the frequency analysis methods described in [6]. The Fast Fourier Transform (FFT) accuracy is affected by the time series length, which should be 2^n , where $n \in \mathbb{R}$, and the time series must contain a whole number of periods of the pulse signal. The raw data contains noise, so these requirements are not satisfied to analyse the magnitude of the spectral coefficient. Consequently, this analysis focuses on the capability to deliver the pulse signal at the desired frequency.

[REDACTED]

Figure 1 — **Fourier Transform of a pulse signal.** **a** $\text{sinc} = \frac{\sin x}{x}$, envelope of the transform. **b** Spectral coefficients of a pulse signal with frequency f_0 and duty cycle, $\delta = 12.5\%$. **c** sinc envelope of the spectral coefficients of a pulse signal with frequency f_0 and $\delta = 50\%$. These images were taken from [1]. **d** Pure pulse signal $f_0 = 120$ Hz, $\delta = 75\%$. **e** Magnitude of the FFT of the pulse signal where the time series doesn't contain a whole number of periods and its length breaks the constraint of being 2^n . This demonstrates the effects on the FFT accuracy when these requirements are not met. In this case, the peak at 120 Hz is higher than that at 240 Hz even though the signal is a pure pulse. The magnitude of the FFT is employed because the FFT involve complex numbers. So, our frequency analysis plots (see Supplementary Figure 3-5) show all the negative values, illustrated in subfigures b and c, as positive.

Sec. **Results**, p. 5: The index finger sensitivity analysis results across frequencies and areas (see Fig. 4a) demonstrate that the sensitivity threshold vary across regions (Kruskal Wallis, $n = 84$, $p = 2.55 \times 10^{-15}$) and frequencies (Kruskal Wallis, $n = 98$, $p = 5.03 \times 10^{-46}$). The results across frequencies are skewed to the right, indicating that the highest stimulus intensity threshold occurs at 2 Hz (two-sided Wilcoxon rank sum test, 1% significance level, $p \leq 1.58 \times 10^{-11}$), with the threshold decreasing as the frequency increases..

Comment 2

Similarly, the assertion regarding multi-point presentation raises some uncertainties, primarily due to the lower point density compared to previous research on multi-point displays. Consequently, asserting the strength of a multi-point (multi-planar) presentation might require further consideration.

Response: We thank the reviewer for their comment. From our understanding, Reviewer 2 seems to be referring to the stimulus density provided by our BAMH system. We like to clarify that our contributions do not concern the stimulus density. Our second contribution, in fact, focuses on the capability of our system to deliver multiplanar, simultaneous stimuli across and within areas. The evaluation was conducted by testing the differentiation of stimuli on the index finger across areas (see our article page 7), as previously requested by Reviewer 2, following the methods described in [9] and the two-point differentiation across fingers (see our article page 6). Both experiments demonstrate that participants' ability to correctly identify the stimuli and their accuracy changes across areas. Therefore, providing different stimuli in all areas of the finger is important.

The paper has now been updated as follows:

Sec. **Results**, p. 8: The spatial resolution of the BAMH system could be maximised to match that of existing systems such as those described by [4, 5].

Matlab code

```
% requires Matlab's Signal Processing Toolbox
clear
close all

Fs=4000 % sampling frequency of the pulse signal is 4000Hz
f=120 % signal frequency of pulse is 120Hz
t=0:1/Fs:2.004; %signal time off with a time step of 1/Fs
X=75*square(2*pi*f*t,75)+75; % generate the pulse signal of amplitude 150

%% Plot the signal in time domain
figure
subplot(1,2,1)
plot(t,X)
ylim([-20 170])
xlim([0 0.06])
ylabel("Amplitude")
xlabel("t (s)")
title("Time domain pulse signal 120Hz 75%")

%% Frequency analysis (as per Matlab documentation 2022a for fft()), Noisy
subplot(1,2,2)
L = length(X);
Y = fft(X);
P2 = abs(Y/L);
P1 = P2(1:L/2+1);
```

```

P1(2:end-1) = 2*P1(2:end-1);
f = Fs/L*(0:(L/2));
plot(f,P1,"LineWidth",3)
title("Single-Sided Amplitude Spectrum of X(t)")
xlabel("f (Hz)")
ylabel("|P1(f)|")
xlim([0 800])

```

References

- [1] Benoit Boulet and L Chartrand. *Fundamentals of signals and systems*. Da Vinci Engineering Press Hingham, MA, 2006.
- [2] Harshal A. Sonar, Aaron P. Gerratt, Stéphanie P. Lacour, and Jamie Paik. Closed-loop haptic feedback control using a self-sensing soft pneumatic actuator skin. *Soft Robotics*, 7(1):22–29, 2020.
- [3] Xinge Yu, Zhaoqian Xie, Yang Yu, Jungyup Lee, Abraham Vazquez-Guardado, Haiwen Luan, Jasper Ruban, Xin Ning, Aadeel Akhtar, Dengfeng Li, et al. Skin-integrated wireless haptic interfaces for virtual and augmented reality. *Nature*, 575(7783):473–479, 2019.
- [4] Takaaki Taniguchi, Sho Sakurai, Takuya Nojima, and Koichi Hirota. Multi-point pressure sensation display using pneumatic actuators. In *Haptics: Science, Technology, and Applications: 11th International Conference, EuroHaptics 2018*, volume 2, pages 58–67, 2018.
- [5] Yusuke Ujitoko, Takaaki Taniguchi, Sho Sakurai, and Koichi Hirota. Development of finger-mounted high-density pin-array haptic display. *IEEE Access*, 8:145107–145114, 2020.
- [6] Yitian Shao, Vincent Hayward, and Yon Visell. Spatial patterns of cutaneous vibration during whole-hand haptic interactions. *Proceedings of the National Academy of Sciences*, 113(15):4188–4193, 2016.
- [7] R. S. Johansson, A. B. Vallbo, and G. Westling. Thresholds of mechanosensitive afferents in the human hand as measured with von frey hairs. *Brain Research*, 184(2):343–351, 1980.
- [8] Roland S Johansson, Ulf Landstro, Ronnie Lundstro, et al. Responses of mechanoreceptive afferent units in the glabrous skin of the human hand to sinusoidal skin displacements. *Brain research*, 244(1):17–25, 1982.
- [9] Aysien Ivanov, Daria Trinitatova, and Dzmitry Tsetserukou. Linkring: a wearable haptic display for delivering multi-contact and multi-modal stimuli at the finger pads. In *Haptics: Sci., Technology, Applicat.: 12th Int. Conf., EuroHaptics 2020*, volume 12, pages 434–441, 2020.

Supplementary Figure 3 : **Raw force data of the BAMH system in the time and frequency domain for pulse stimulus, with $\delta = 75\%$, between 0 Hz and 50 Hz.** The frequency domain plots demonstrate the BAMH system's capability to deliver pulse stimulus by exhibiting the spectral coefficients at the frequencies that are a multiple of the stimulus frequency.

Supplementary Figure 4 : **Raw force data of the BAMH system in the time and frequency domain for pulse stimulus, with $\delta = 75\%$, between 55 Hz and 100 Hz.** The frequency domain plots demonstrate the BAMH system's capability to deliver pulse stimulus by exhibiting the spectral coefficients at the frequencies that are a multiple of the stimulus frequency.

Supplementary Figure 5 : **Raw force data of the BAMH system in the time and frequency domain for pulse stimulus, with $\delta = 75\%$, between 105 Hz and 130 Hz.** The frequency domain plots demonstrate the BAMH system's capability to deliver pulse stimulus by exhibiting the spectral coefficients at the frequencies that are a multiple of the stimulus frequency.

REVIEWERS' COMMENTS

Reviewer #2 (Remarks to the Author):

I thank the authors for their detailed explanation. I now understand my misunderstanding in the previous round and sincerely apologize for any confusion caused. My concerns have been thoroughly resolved.

**Bioinspired adaptable multiplanar
mechano-vibrotactile haptic system
(NCOMMS-23-44661C-Z): Comments and
answers**

DBPR

Many thanks to the reviewers for acknowledging our effort in revising the paper and addressing your constructive comments thoroughly. We also express our gratitude for the time dedicated to reviewing our responses.